# Probability Consistency in Large Language Models: Theoretical Foundations Meet Empirical Discrepancies

## Abstract

Can autoregressive large language models (LLMs) learn consistent probability distributions when trained on sequences in different token orders? We prove formally that for any well-defined probability distribution, sequence perplexity is invariant under any factorization, including forward, backward, or arbitrary permutations. This result establishes a rigorous theoretical foundation for studying how LLMs learn from data and defines principled protocols for empirical evaluation. Applying these protocols, we show that prior studies examining ordering effects suffer from critical methodological flaws. We retrain GPT-2 models across forward, backward, and arbitrary permuted orders on scientific text. We find systematic deviations from theoretical invariance across all orderings with arbitrary permutations strongly deviating from both forward and backward models, which largely (but not completely) agreed with one another. Deviations were traceable to differences in self-attention, reflecting positional and locality biases in processing. Our theoretical and empirical results provide novel avenues for understanding positional biases in LLMs and suggest methods for detecting when LLMs' probability distributions are inconsistent and therefore untrustworthy.

## 1 Introduction

Transformer-based (Vaswani et al., 2023) decoder-only large language models (LLMs) owe their success largely to a scalable architecture and an autoregressive loss function grounded in probability theory. These models approximate the conditional probability distribution over a sequence of tokens, denoted as $P(X_t \mid X_{1:t-1})$, where $X_t$ is the $t$-th token in a sequence $X_1, X_2, \ldots, X_t$, and $X_{1:t-1} = X_1, \ldots, X_{t-1}$ represents the prior context. The training objective minimizes the negative log-likelihood loss: $\mathcal{L} = -\frac{1}{n} \sum_{t=1}^{n} \ln P(X_t \mid X_{1:t-1})$, where $n$ is the sequence length, encouraging the model to assign high probabilities to correct tokens given their context. This leverages vast text corpora to learn a probability distribution over sequences in a vocabulary $\mathcal{V}$. A valid probability distribution belongs to the set $\mathcal{P}$, satisfying $P(X_1, \ldots, X_n) \geq 0$ for any sequence $X_1, \ldots, X_n$, and normalizing such that: $\sum_{(X_1, \ldots, X_n) \in \mathcal{V}^n} P(X_1, \ldots, X_n) = 1$. The chain rule ensures consistency in factorization: $P(X_1, \ldots, X_n) = \prod_{t=1}^{n} P(X_t \mid X_{1:t-1})$.

The theoretical foundation of LLMs prompts a critical question: does a formal basis exist to guarantee that probability estimates remain consistent across models trained on the same data but with different token orderings, such as forward $(X_1, \ldots, X_n)$ versus reverse $(X_n, \ldots, X_1)$? Do practical LLMs, which approximate these probabilities, uphold this consistency? If so, under what conditions? If not, what causes the discrepancies? Inconsistencies in probability estimates may arise from practical limitations, such as learning order and data complexity (Lampinen et al., 2024; Zucchet et al., 2025), gradient-based optimization, architectural biases and constraints (Liu et al., 2023; Latuske et al., 2015; Wu et al., 2025; Bondarenko et al., 2023; Gu et al., 2025) or numerical precision limitations (Barbero et al., 2024). These deviations could reflect gaps in our theoretical understanding of LLMs, potentially contributing to issues like hallucinations (see Huang et al. 2025 for a review) or unfaithful or inconsistent reasoning (Lanham et al., 2023; Lindsey et al., 2025; Chen et al., 2025). Investigating whether a theoretical framework ensures probability consistency and identifying the conditions for its validity is crucial for understanding the learning dynamics of these models and how to best evaluate them.

In this contribution, we provide a mathematical proof addressing this theoretical consistency. We demonstrate rigorously using the chain rule of probability that sequence perplexity, which is fundamentally determined by the joint probability $P(X_1, \ldots, X_n)$, is theoretically invariant to the order of factorization. This means that calculating perplexity using forward conditionals, backward conditionals, or indeed any fixed permutation of the tokens, must yield the exact same result for a true probability distribution. While prior work has suggested potential equivalence between forward and backward processing (Zhang et al., 2018; Papadopoulos et al., 2024; Zhang et al., 2025), our proof formalizes this and generalizes it to arbitrary factorizations, establishing a clear theoretical benchmark against which practical model behavior can be assessed (detailed in Section 2). Though the chain rule itself is well-known, our application to perplexity invariance under training order (i.e., factorization) is novel. This theoretical invariance provides a measuring stick to evaluate deviations in practice. Providing this theoretical framework and using it to examine empirical deviations is our primary contribution.

Despite the theoretical expectation of perplexity equivalence across token orderings, empirical studies have reported discrepancies. For instance, Kallini et al. (2024) found autoregressive transformers favor natural language order. Similarly, Papadopoulos et al. (2024) and Luo et al. (2024a) observed lower perplexities in forward-trained models. In contrast, Yu et al. (2025) noted inconsistent perplexity differences across text domains, while Zhang et al. (2025) corroborated lower forward perplexities, aligning with Papadopoulos et al. (2024). Additionally, Luo et al. (2024a) and Zhang et al. (2025) reported superior performance of backward-trained models in multiple-choice tasks.

However, the aforementioned studies do not speak to the theoretical equivalence of forward and backward sequences due to methodological choices in their experimental setups. Previous efforts hinge on comparing models trained on different token orderings. For such a comparison to be informative in regards to equivalence, the token ordering must be the sole variable that changes between models. However, these studies inadvertently analyzed distinct text sequences instead of different factorizations of the same sequence due to flaws stem from errors such as omitting consistent begin-of-sequence tokens (Kallini et al., 2024; Luo et al., 2024a; Yu et al., 2025), retraining tokenizers for reversed text (Papadopoulos et al., 2024; Luo et al., 2024a), and conflating logical reversal with strict token sequence reversal (Zhang et al., 2025; Papadopoulos et al., 2024) (Table 1; see Appendix A for details). When studies fail to maintain consistent BOS tokens and tokenizers across factorizations, the resulting models learn to approximate probabilities of entirely different sequences, not different orderings of the same sequence. This means the forward and backward models no longer capture the same underlying joint probability, which complicates comparing performance. If the research goal is to compare the processing of sequences that vary only in order, then it is critical to avoid these methodological pitfalls. We outline a framework to help researchers avoid these pitfalls, which should aid in the interpretation of results and open the door for new applications.

Our proof serves to address the errors in prior studies and establishes precise protocols for empirically investigating learning inconsistencies in LLMs. These protocols ensure proper handling of special tokens, consistent tokenization strategies, and strict alignment of training orderings. To demonstrate their efficacy, we revisit the experimental setup of Luo et al. (2024a) and conduct theory-aligned training and evaluations in the neuroscience domain. By formalizing the theoretical expectation of perplexity invariance, our proof provides the necessary foundation for designing valid experimental comparisons and interpreting discrepancies in LLM behavior.

To precisely follow the proof and ensure valid sequence perplexity comparisons, we trained 27 GPT-2 models (Radford et al., 2019) at three scales (124M, 355M, 774M) on twenty years of neuroscience publications, using forward, backward, and permuted token orderings. We adhered to strict protocols: each sequence starts with a begin-of-sequence (BOS) token, employs a tokenizer trained solely on forward text across all factorizations, and applies token permutations within the context window. Models with different token orderings use identical data sequences in the same order, differing only in token arrangement within the context (see Appendix D for details). We evaluated model differences by analyzing consistencies of perplexity, attention strategies, representational alignment and accuracy on BrainBench (Luo et al., 2024b), a neuroscientist-curated benchmark that tests models' ability to distinguish original experimental results from subtly altered versions.

To foreshadow key results, we find forward- and backward-trained models exhibit broadly similar perplexity, attention patterns, and downstream performance, though small but systematic discrepancies emerge, contrary to our theoretical predictions. These discrepancies are greatly amplified in models trained on permuted text, where we trace inconsistencies to positional biases in self-attention. Our analysis reveals that causal self-attention exhibits both locality and long-range biases, extending prior observations of early-position biases. We further clarify that these biases arise not only from model architecture but also from the structure of the training data, beyond current explanations of attention biases (Liu et al., 2023; Xiao et al., 2024). While prior work has noted the theoretical equivalence of joint probabilities within specific settings, our study strengthens the empirical foundation with an explicit mathematical proof. Our work not only corrects errors in earlier research, but more importantly clarifies a key line of inquiry into probability consistency in LLMs, bridges the previously disconnected domains of probability learning and architectural biases and illuminates some critical aspects of the way LLMs learn from data.

Table 1: Methodological Deviations from Theoretical Proof in Recent Studies

| Deviation Category | Study |
| --- | --- |
| Missing begin-of-sequence and/or final tokens | Kallini et al. (2024), Luo et al. (2024a) |
| | Yu et al. (2025) |
| Retraining tokenizer on backward text | Papadopoulos et al. (2024) |
| | Luo et al. (2024a) |
| Mixing logical reversal with token reversal | Zhang et al. (2025) |
| | Papadopoulos et al. (2024) |

## 2 Proof of Perplexity Equivalence in Sequences of Arbitrary Factorizations

The perplexity of a sequence of tokens is typically calculated by factorizing the joint probability into conditional probabilities in a forward direction, which allows for practical computation using language models that predict next tokens given previous context. While this conditional decomposition suggests a directional nature, we show that sequence perplexity fundamentally measures the joint probability of the entire sequence. Calculations of sequences factorized in different orders are simply different paths to recover this same joint probability, guaranteed equal by the chain rule of probability.

For practical autoregressive language models, which predict each token based on prior context, we introduce a begin-of-sequence (BOS) token, $X_0$, to provide the initial context needed for the model to predict the first token. While not strictly necessary in a general mathematical context, $X_0$ ensures the first conditional probability is well-defined in this setting. We define $P(X_0) = 1$, as it has no prior context. Let $X_1, X_2, \ldots, X_n$ be a sequence of $n$ tokens following $X_0$. Let $\sigma$ be a permutation of the indices $\{1, 2, \ldots, n\}$, defining a reordering: $X_{\sigma(1)}, X_{\sigma(2)}, \ldots, X_{\sigma(n)}$.

The perplexity for this ordering is defined as

$$\text{PP}_\sigma = \exp\left(-\frac{1}{n} \sum_{i=1}^{n} \ln P(X_{\sigma(i)} | X_0, X_{\sigma(1)}, \ldots, X_{\sigma(i-1)})\right).$$

By the chain rule of probability,

$$P(X_{\sigma(i)} | X_0, X_{\sigma(1)}, \ldots, X_{\sigma(i-1)}) = \frac{P(X_0, X_{\sigma(1)}, \ldots, X_{\sigma(i)})}{P(X_0, X_{\sigma(1)}, \ldots, X_{\sigma(i-1)})},$$

we rewrite the perplexity as

$$\text{PP}_\sigma = \exp\left(-\frac{1}{n} \sum_{i=1}^{n} \ln \frac{P(X_0, X_{\sigma(1)}, \ldots, X_{\sigma(i)})}{P(X_0, X_{\sigma(1)}, \ldots, X_{\sigma(i-1)})}\right)$$

$$= \exp\left(-\frac{1}{n} \sum_{i=1}^{n} \left[\ln P(X_0, X_{\sigma(1)}, \ldots, X_{\sigma(i)}) - \ln P(X_0, X_{\sigma(1)}, \ldots, X_{\sigma(i-1)})\right]\right).$$

This sum telescopes:

$$\sum_{i=1}^{n} \left[ \ln P(X_0, X_{\sigma(1)}, \ldots, X_{\sigma(i)}) - \ln P(X_0, X_{\sigma(1)}, \ldots, X_{\sigma(i-1)}) \right] = \ln P(X_0, X_{\sigma(1)}, \ldots, X_{\sigma(n)}) - \ln P(X_0),$$

since $P(X_0) = 1$, so $\ln P(X_0) = 0$, yielding

$$\mathrm{PP}_\sigma = \exp\left( -\frac{1}{n} \ln P(X_0, X_{\sigma(1)}, \ldots, X_{\sigma(n)}) \right).$$

Since $\{X_{\sigma(1)}, \ldots, X_{\sigma(n)}\}$ is a permutation of $\{X_1, \ldots, X_n\}$, the joint probability is unchanged:

$$P(X_0, X_{\sigma(1)}, \ldots, X_{\sigma(n)}) = P(X_0, X_1, \ldots, X_n).$$

Thus,

$$\mathrm{PP}_\sigma = \exp\left( -\frac{1}{n} \ln P(X_0, X_1, \ldots, X_n) \right). \tag{1}$$

This general result encompasses the equivalence of perplexities of forward and backward factorized sequences as special cases. We provide the corresponding proof in Appendix B. We also provide intuitive examples illustrating this equivalence in Appendix C.

**Implications for Language Model Training**  The theoretical proof demonstrates that, in language model training, the choice of token prediction order—left-to-right, right-to-left, or any fixed permutation—yields the same theoretical perplexity, provided the model accurately captures the conditional probabilities. However, each autoregressive LLM captures only a single factorization of the training data during training and cannot be meaningfully evaluated on different token orderings than those it was trained on. To empirically verify this theoretical equivalence, sibling models must be trained on different factorizations. For example, a forward-trained model does not explicitly model conditionals from backward factorizations, and approximating intermediate conditionals of another factorization from a single trained model via sampling is computationally infeasible for large text corpora. Therefore, for each order considered, an LLM must be trained and evaluated on its specific token ordering to properly assess the factorization of the joint probability distribution it has learned.

## 3  Experiments

To evaluate whether LLMs approximate consistent probabilities across token orderings, we train 27 GPT-2 models (Radford et al., 2019) from scratch at three scales (124M, 355M, 774M parameters) on a 1.3-billion-token corpus of neuroscience publications spanning 20 years. Prior work showed that models of this size, trained on the same data, matched or exceeded expert performance on a neuroscience benchmark (Luo et al., 2024c). Here, the models were trained with forward, backward, and permuted token orderings, each with three different initializations. To align with the theoretical framework, all sequences include a begin-of-sequence (BOS) token and fully span the model's context window. A new tokenizer, trained on the same neuroscience corpus using forward factorization, ensures all models process identical data in consistent orders. All models are trained for five epochs to convergence. We assess model consistency through perplexity (Sec. 3.1), attention strategies (Sec. 3.2), representational alignment (Sec. 3.3) and accuracy on BrainBench (Sec. 3.4; Luo et al. 2024b), a neuroscientist-curated benchmark that evaluates models' ability to distinguish original experimental results from subtly altered versions (see Appendix D, E for details).

### 3.1  Perplexity Differences Across Sequence Factorizations

Forward and backward-trained models show highly similar perplexities across validation set text sequences ($N = 9,413$; Fig. 1), with Pearson correlation coefficients greater than 0.99 (Table 2). However, forward-trained models consistently exhibit lower perplexity than backward-trained models, with the gap widening

as model size increases (Table 2). Models trained on permuted text exhibit significantly higher perplexities, diverging markedly from theoretical expectations (Fig. 1; Table 2). For perplexity differences on the training set, the complete training and validation losses and initialization differences, see Appendix F (Fig. S.1, S.2; Table S.1, S.2).

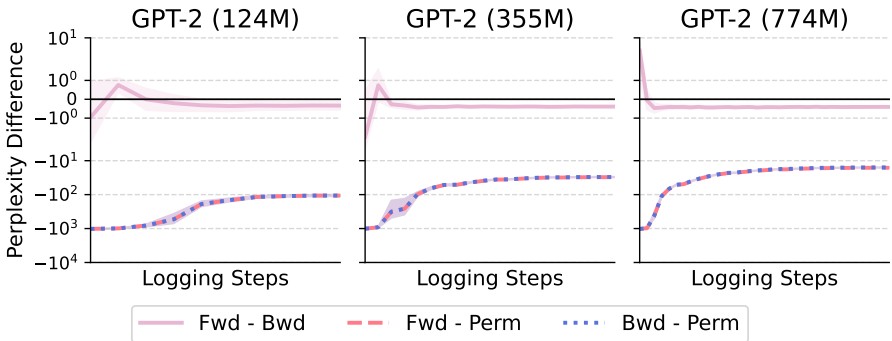

Figure 1: **Average validation perplexity differences across across model sizes and training directions.** Forward and backward text training yields similar perplexities, though forward models consistently achieve lower values (difference below zero). This gap widens slightly with model size. Permuted text training yields much higher perplexity than both forward and backward models, with similar differences to each, causing the curves to overlap. Shaded regions indicate one standard deviation over the mean across three random initializations.

Table 2: Comparison of Statistical Metrics Across GPT-2 Model and Directions Across Three Initializations ($***$ indicates $p < 0.001$)

| Model | Direction | Pearson $r$ | $t$-stat | Cohen's $d$ |
|---|---|---|---|---|
| 124M | Fwd vs Bwd | $0.995 \pm 0.000^{***}$ | $-18.486 \pm 8.795^{***}$ | $0.191 \pm 0.091$ |
| | Fwd vs Perm | $0.889 \pm 0.001^{***}$ | $-230.581 \pm 2.081^{***}$ | $2.377 \pm 0.021$ |
| | Bwd vs Perm | $0.888 \pm 0.003^{***}$ | $-230.148 \pm 2.022^{***}$ | $2.372 \pm 0.021$ |
| 355M | Fwd vs Bwd | $0.995 \pm 0.000^{***}$ | $-40.788 \pm 1.958^{***}$ | $0.420 \pm 0.020$ |
| | Fwd vs Perm | $0.930 \pm 0.004^{***}$ | $-199.889 \pm 1.077^{***}$ | $2.060 \pm 0.011$ |
| | Bwd vs Perm | $0.930 \pm 0.004^{***}$ | $-198.720 \pm 1.040^{***}$ | $2.048 \pm 0.011$ |
| 774M | Fwd vs Bwd | $0.995 \pm 0.000^{***}$ | $-64.484 \pm 1.459^{***}$ | $0.665 \pm 0.015$ |
| | Fwd vs Perm | $0.957 \pm 0.001^{***}$ | $-208.686 \pm 0.935^{***}$ | $2.151 \pm 0.010$ |
| | Bwd vs Perm | $0.958 \pm 0.001^{***}$ | $-207.833 \pm 0.675^{***}$ | $2.142 \pm 0.007$ |

### 3.2 Tracing Ordering Effects to Self-Attention Biases

To understand the sources of the perplexity differences, we analyze the attention patterns in models trained on differently ordered text sequences. If self-attention inherently biases toward certain token positions, these positional preferences could cause discrepancies in perplexity across different factorizations.

**Token Ordering Shapes Attention Entropy Across Models**  We investigate variations in attention weight distributions across models by analyzing their entropy. Let $A^{(l,h)} \in \mathbb{R}^{T \times T}$ denote the attention weight matrix for a specific head $h$ in layer $l$, where $T$ is the sequence length. Each entry $A_{ij}^{(l,h)}$ represents the attention probabilities from token $i$ to token $j$. For each token $i$, its attention distribution is defined over the preceding tokens and itself ($j \leq i$), denoted $a_i^{(l,h)} = (A_{i1}^{(l,h)}, A_{i2}^{(l,h)}, \ldots, A_{ii}^{(l,h)})$ with $\sum_{j=1}^{i} A_{ij}^{(l,h)} = 1$. The entropy for the attention distribution of token $i$ is then $H(a_i^{(l,h)}) = -\sum_{j=1}^{i} A_{ij}^{(l,h)} \ln A_{ij}^{(l,h)}$. For $i = 1$, we define $H(a_1^{(l,h)}) = 0$. To account for varying context sizes, we normalize this entropy by the maximum

possible entropy for a distribution over $i$ tokens, which corresponds to a uniform distribution ($p_j = 1/i$ for $j = 1, \ldots, i$). The maximum entropy is: $H_{\max}(i) = -\sum_{j=1}^{i} \frac{1}{i} \ln \frac{1}{i} = \ln i$. For $i = 1$, $H_{\max}(1) = \ln 1 = 0$. The normalized entropy for token $i$ is then: $\hat{H}(a_i^{(l,h)}) = \frac{H(a_i^{(l,h)})}{H_{\max}(i)} = \frac{H(a_i^{(l,h)})}{\ln i}$ for $i > 1$, and $\hat{H}(a_1^{(l,h)}) = 0$.

We compute the normalized entropy by averaging across all heads and 64 text sequences per context size, resulted in a scalar value representing the variation of attention for each context size for each model layer. Sequences are sampled from the validation split of twenty years of neuroscience publications, each fully spanning the model's context window with a BOS token prepended.

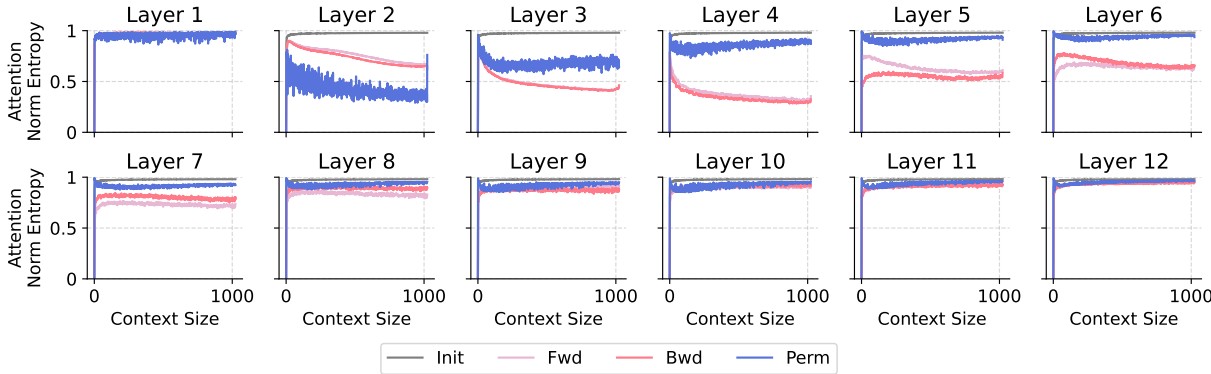

Figure 2: **Attention entropy across three data orders (GPT-2 124M).** Normalized attention entropy ($min = 0, max = 1$) is measured across layers averaged over heads and sampled text sequences for varying context sizes. Models trained on forward, backward, and permuted token orders show distinct patterns despite using the same data. Forward and backward models exhibit similar trends, with larger differences at early layers. The model trained with permuted token order displays substantially higher and more divergent entropy, particularly in early to middle layers, suggesting distinct learning dynamics driven by unnatural local and long-range dependencies. Models at initialization (Init) are shown for reference and display near-maximal entropy. All test sequences fully span the context window each prepended with a BOS token.

We observe that models trained on forward and backward token orders show similar entropy patterns, while the model trained on permuted token order exhibits much higher and more divergent entropy, especially in early to middle layers (Fig. 2). We attribute this to disrupted grammatical dependencies caused by consistent token permutation. Although attention entropy across models tends to converge in the later layers, earlier discrepancies likely cascade, leading to overall divergence in perplexity despite theoretical equivalence. Results for different initializations and model sizes are in the Appendix F.

**Attention Weights Are Biased by Token Positions**  We complement our entropy analysis by examining how attention weights vary across token pairs within a sequence based on the distance of their relative positions. To understand the relative importance of tokens within this context, we compute the rank of each attention weight $A_{ij}^{(l,h)}$ within the set $a_i^{(l,h)}$. Let $R(A_{ij}^{(l,h)}|a_i^{(l,h)})$ denote the rank of $A_{ij}^{(l,h)}$ among the $i$ weights in $a_i^{(l,h)}$, where ranks range from 0 (for the smallest weight) to $i - 1$ (for the largest weight). We normalize these ranks to a $[0,1]$ scale to account for the varying context size $i$. The normalized rank for the attention weight $A_{ij}^{(l,h)}$ is $\hat{R}_{ij}^{(l,h)} = \frac{R(A_{ij}^{(l,h)}|a_i^{(l,h)})}{i-1}$ for $i > 1$ and $\hat{R}_{11}^{(l,h)} = 0$.

We analyze how normalized ranks relate to the distance $d = |i - j|$ between the query token $i$ and the key token $j$, for each possible distance $d \in \{0, 1, \ldots, T - 1\}$. We aggregate normalized ranks by averaging them across all valid token pairs $(i, j)$ such that $j \leq i$ and $|i - j| = d$, across all heads in a layer, and across 64 randomly sampled text sequences fully spanning the context window with BOS prepended) from the same neuroscience dataset.

We observe in Fig. 3 that compared to models at initialization (Init), both forward and backward trained models exhibit strong biases toward both adjacent tokens and those at the maximum context length, with

the strength of bias varying across layers. In contrast, the model trained on permuted text shows a distinct trend, with positional bias generally decreasing as token distance increases across most layers.

We confirm that biases toward nearby tokens and those at maximal distances are a general phenomenon across pre-trained models (e.g., GPT-2, Pythia, and Llama-2) when evaluated on general text corpora like The Pile (Gao et al., 2020) (see Appendix F; Fig. S.3, S.4, S.5). While the exact form of the bias varies, we attribute these differences to factors such as the greater diversity and scale of the pre-training data, which likely support more nuanced learning across token distances. Full results across different initializations and model sizes are provided in Appendix F.

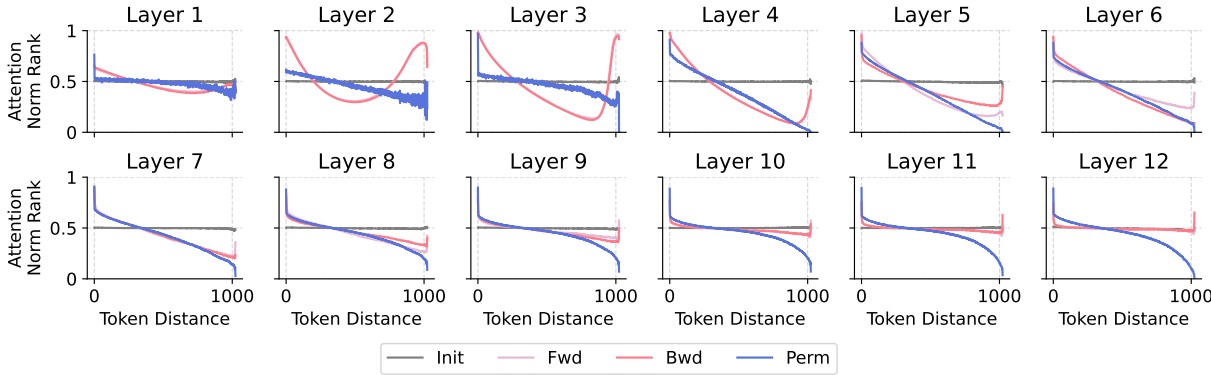

Figure 3: **Positional bias in self-attention varies with training directions and layers (GPT-2 124M).** Normalized attention rank ($min = 0, max = 1$) is plotted as a function of token distance within the context, averaged across heads, sampled sequences, and layers. Compared to models at initialization (Init), forward (Fwd) and backward (Bwd) trained models show strong positional biases toward both nearby tokens and tokens at maximal distance, with the degree of bias varying across layers. In contrast, the model trained on permuted text (Perm) displays distinct patterns, with positional bias generally decreasing as token distance increases across most layers. All test sequences fully span the context window each prepended with a BOS token.

### 3.3 Representational Divergence Across Token Orderings

Having established differences in attention strategies, we trace out how such differences across models affect downstream representational semantics, focusing on how identical text sequences are represented in different models. Specifically, we compare hidden state representations from each self-attention block of same-sized models (Kornblith et al., 2019).

For layer $l$, hidden states $H^{(l)} \in \mathbb{R}^{T \times D}$ ($T$: sequence length, $D$: hidden dimension) are extracted from 64 randomly sampled sequences. Hidden states are reordered to forward sequence order. Representational Dissimilarity Matrices (RDMs) are computed as $\text{RDM}_{i,j}^{(l)} = 1 - \frac{H_i^{(l)} \cdot H_j^{(l)\top}}{||H_i^{(l)}|| \cdot ||H_j^{(l)}||}$. For two models, RSA is the Spearman's rank correlation coefficient of the upper triangular elements of their RDMs. RSA is aggregated by averaging across batches and sampled sequences per layer.

We observe that as layers advance, representational alignment decreases across all direction pairs. Semantic alignment is significantly higher between forward- and backward-trained models than with permuted models. This mirrors our earlier observations on attention strategies, where forward and backward models exhibited more similar attention patterns to each other than to the permuted model (Fig. 2, 3), underscoring fundamental differences in learning dynamics.

### 3.4 BrainBench Evaluations

To provide an external performance measure of evaluation, we extend our analysis beyond perplexity inconsistencies to include downstream benchmark performance. We assess both forward- and backward-trained

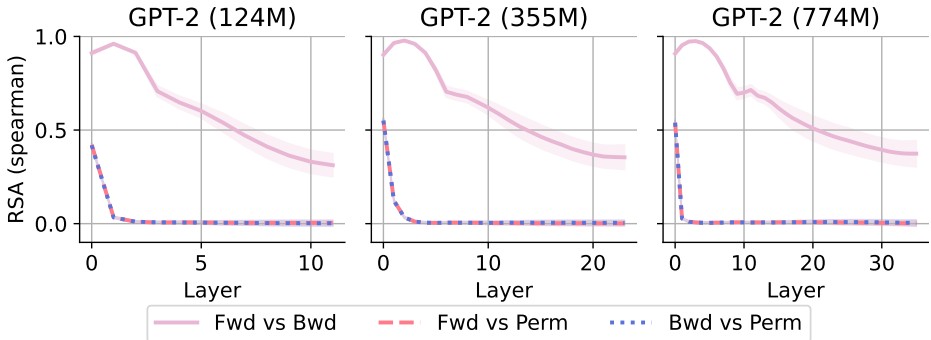

Figure 4: **Representational similarities across training directions.** Forward- and backward-trained models show higher representational similarity to each other than to the model trained on permuted text. Across all comparisons, similarity declines in deeper layers, with the permuted model's representations becoming increasingly orthogonal (toward zero correlation) to the others, indicating a diverging semantic structure from models trained on forward and backward orderings.

models on BrainBench, a recognized benchmark task requiring differentiation of original experimental results from altered versions, compared against human-expert performance (details in Appendix E), as utilized by Luo et al. (2024a). This evaluation is critical, as our work rigorously follows specified protocols to ensure perfect comparability between forward- and backward-trained models, correcting experimental errors in Luo et al. (2024b) and enabling a robust reassessment of their results and interpretations. We exclude permuted-text models, which converged during training (Fig. S.2), but cannot be evaluated on BrainBench, as its items do not always span the full context window and become invalid under the training permutation.

**Forward and Backward-Trained Models Perform Similarly on BrainBench**  We made similar observation as Luo et al. (2024a) that GPT-2 models pre-trained on neuroscience rival or exceed human expert performance (Fig. S.24). As model size increased, both forward and backward-trained models improved in BrainBench performance ($F = 133.397, p = 1.53e - 08$). However, we found no advantage in backward-trained models over forward-trained ones ($F = 1.868, p = 0.1933$) and the interaction between direction and model size was not significant ($F = 4.035, p = 0.0643$; Appendix E).

**Forward and Backward-Trained Models Align with Human Judgements Similarly**  Following Luo et al. (2024a), we examined whether forward-trained and backward-trained models, alongside human experts, identified the same BrainBench items as difficult. Difficulty was quantified for humans by calculating mean accuracy across 200 test cases and for GPT-2 models by computing signed perplexity differences between incorrect and correct abstracts per test case.

Consistent with Luo et al. (2024a), we found that regardless of training direction, model judgments correlated more strongly with each other ($M = 0.74, SD = 0.08$) than with human judgments ($M = 0.11, SD = 0.04$; Fig. S.25, S.26). In contrast to Luo et al. (2024a), however, correlations between forward-trained ($M = 0.12, SD = 0.03$) and backward-trained ($M = 0.10, SD = 0.04$) model judgments and human expert judgments were comparable ($t(8) = 1.009, p = 0.342$; Fig. S.25). This differs from Luo et al. (2024a), who reported significantly lower alignment for backward-trained models with human judgments. This difference stems from corrections we made to their experimental setup: we ensured that forward- and backward-trained models shared identical initialization conditions, were trained on exactly the same data sequences fully spanning the context window with a BOS token prepended, and used consistent tokenizations. This divergence in findings has important implications for model-human alignment and the design of effective model-human teaming systems (Yáñez et al., 2025).

## 4  Discussion

In this contribution, we formally prove that sequence perplexity is theoretically invariant to factorization order under the chain rule of probability, establishing a mathematical benchmark for evaluating how consistent LLMs are at approximating conditional probabilities. Empirically, however, models trained on forward, backward, and permuted token orders exhibit systematic deviations from this equivalence. Forward and backward models achieve similar (but not identical) perplexities, with forward models consistently outperforming backward models, while permuted training yields significantly higher losses. Attention analysis reveals that forward/backward models develop strong positional biases—favoring adjacent and long-range tokens—whereas permuted models exhibit distinct attention patterns. Representational alignment is higher between forward and backward models but degrades with depth, diverging sharply for permuted models. Finally, evaluation on the BrainBench benchmark shows that both forward and backward models match human expert performance but align poorly with human judgment patterns, contradicting prior claims about the inferior alignment of backward models. Together, these findings reveal how architectural and data-driven biases disrupt theoretical equivalence and shape LLM learning dynamics.

While our proof establishes theoretical equivalence across factorizations, it may seem counterintuitive that natural language sequences exhibit identical perplexity regardless of token ordering. This mathematical property holds for any sequence of tokens, yet no natural language on earth resembles the permuted orders we considered. There are likely several reasons for why natural languages do not manifest such structures. For instance, people, like LLMs, have a locality bias that may arise from working memory constraints. People process language in real-time, not by calculating the perplexity of an entire passage. Certain language structures, such as ones that balance changes in uncertainty throughout the sequence, should be easier for humans to process. Finally, our analyses do not involve generating sequences. Instead, we focus on evaluating existing sequences using perplexity. Natural languages are likely selected to not only favor perception, but also production (i.e., generation).

While we trained GPT-2 models on backward and permuted text sequences to learn different factorizations of joint probability and demonstrate discrepancies from theoretical equivalence, one might question whether forward-trained models alone could recover different factorizations. In principle, we could sample from forward models' conditional probabilities to construct joint probabilities factorized in different orders. However, this approach is computationally infeasible given the enormous sampling space across all possible text token combinations. Therefore, training complementary models on various token orderings represents the most practical approach for this investigation.

Understanding the discrepancies between the way models learn in practice and how they might be expected to learn in theory has significant implications. Although we identified errors in prior work that need correction, we acknowledge their contributions in revealing interesting implications of such asymmetries. For instance, Yu et al. (2025) suggested alternative sequence learning directions based on different factorizations could reveal unique insights about data distributions and potentially indicate data quality. Papadopoulos et al. (2024) and Zhang et al. (2025) proposed theoretical frameworks explaining why discrepancies occur, informing our understanding of language model learning dynamics. We also see connections to Wu et al. (2025), who identified position bias in self-attention—leading us to suspect this bias may contribute to asymmetric perplexities that challenge theoretical equivalence.

Beyond theoretical considerations, researchers have found practical utility in backward training approaches. Pfau et al. (2023) trained backward models to identify adversarial prompts for detecting toxic responses. Zhang et al. (2018) optimized bidirectional agreement at sequence-level in machine translation. Nguyen et al. (2023) jointly optimized forward and backward autoregressive language models, maximizing token-level agreement to provide denser supervision signals.

Our findings on causal self-attention biases connect with and extend prior work, such as attention sinks (Xiao et al., 2024; Yu et al., 2024; Gu et al., 2025) and the lost-in-the-middle phenomenon (Liu et al., 2023). One observed bias is consistent with attention sinks (Xiao et al., 2024), where initial tokens dominate attention. This manifests in our analysis (Fig. 3) as high attention from final tokens towards initial tokens, reflected by the increased normalized attention at the far right of the axis. Additionally, we identify a pronounced

locality bias, consistent with prior studies on vanilla transformers (Qin et al., 2022) and bidirectional models like BERT (Clark et al., 2019; Kovaleva et al., 2019), where attention tends to concentrate on nearby tokens.

Notably, our work highlights a more comprehensive view of causal self-attention biases across all token distances irrespective of context size, unlike the focus on only the first few tokens in Xiao et al. (2024) (cf. Yu et al. 2024). By using normalized ranked attention, we avoid the dilution of raw probabilities caused by softmax in longer contexts, in contrast to Xiao et al. (2024)'s reliance on raw probabilities, which treats attention as a limited resource. While Xiao et al. (2024) attributed sinks to early token visibility under masked self-attention, our results with permuted models suggest sinks might not arise under different data factorizations.

Furthermore, our findings offer an empirical explanation for the lost-in-the-middle effect (Liu et al., 2023), where information at the start or end of the context is retrieved best. The strong attention observed for both local tokens and long-range (initial-final) token pairs (Fig. 3) directly accounts for why intermediate information might receive less focus and be harder to retrieve.

The attention biases we systematically characterize directly drive the theory-practice gap we identified, with concrete consequences: models fail to retrieve mid-context information reliably (affecting retrieval-augmented generation) and exhibit unpredictable reasoning behavior. This makes understanding these biases critical for building interpretable and trustworthy systems.

In hindsight, it may seem unsurprising that architectural biases toward specific token positions lead to discrepancies in the theoretical equivalence we established. However, prior work on positional biases and forward-backward inconsistencies developed largely independently, and the connections between them remained unexplored. While Kallini et al. (2024) speculated that locality bias might contribute to forward-backward differences, a systematic analysis linking attention patterns to these discrepancies had not been conducted until now.

Studying the discrepancies between theoretical expectations and practical implementations offers valuable insights into learning dynamics and biases in transformer architectures, which dominate today's AI landscape. We suspect that deviations from proper probability distributions may contribute to thorny problems like hallucination and unpredictable out-of-distribution behaviors in LLMs—issues that fundamentally undermine model interpretability and trustworthiness in applied systems. These deviations could serve as diagnostic metrics for identifying when and why models deviate from principled behavior, enabling better detection and mitigation of unexpected behaviors. One potential mitigation strategy could involve training two or more sibling models simultaneously with a shared objective to minimize divergence in perplexity, encouraging more consistent and stable learning. While this may evoke bidirectional models like BERT (Devlin et al., 2019), it is worth noting that BERT achieves symmetric probabilities by design due to bi-directional attention, which does not optimize joint probabilities—since masked tokens are predicted independently.

Our work, which found empirical differences in perplexity across training orders (i.e., factorizations) despite all orders having have the same perplexity in the abstract, opens up several important research directions. A central question concerns scalability: how do these biases manifest in real-world scenarios beyond controlled experimental settings? We are also curious whether other architectures and models employing different positional embeddings exhibit similar patterns, and whether developing better benchmarks specifically designed to evaluate such divergences might provide clearer signals for practitioners. Understanding the downstream implications of these divergences for model behavior and performance remains largely unexplored. Additionally, we wonder whether making invariance an explicit training constraint could improve model performance by making estimated conditional probabilities mutually consistent. While these connections and proposed strategies are promising, they require extensive investigation beyond the scope of this work. Our current contribution focuses on laying the theoretical groundwork and establishing experimental design best practices as a foundation for such future efforts. Future research in this area holds significant promise for developing more reliable and interpretable LLMs, with the potential to transform how we diagnose and mitigate model limitations.

**Broader Impact Statement**

As a theoretical contribution, this work establishes a solid foundation and practical protocols for theory-aligned model training. Although our empirical experiments are extensive, they do not extend to training multi-billion-parameter, state-of-the-art LLMs on diverse datasets. We envision our theoretical framework serving as a benchmark for such models at greater scale. Notably, prior work, despite identified errors, has already observed inconsistencies in larger models for various training domains, lending support to our premise. Further, in our current work, we observe consistent positional biases in larger LLMs trained on general text, suggesting that training on data of varying factorizations would likely reveal corresponding inconsistencies. Future work could involve training such models with varied factorizations, benchmarking their inconsistencies against theoretical predictions, and investigating how these inconsistencies manifest across diverse LLM applications — ultimately supporting the development of more accountable and interpretable LLM systems with positive societal impact.

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

## A  Existing Work's Misalignment with Theory

Studies (Table 1) which have noted discrepancies of joint probabilities of forward and backward factorizations of the same text sequence often deviate from the theoretical premise due to subtle experimental design or data handling issues. Consequently, when they compared joint probabilities of sequences with different factorization orders, they were in fact inadvertently analyzing distinct text sequences.

These deviations from the theoretical setup weaken confidence in empirical findings, highlighting the need for rigorous adherence to the proof's conditions to ensure valid interpretations.

## A.1 Missing Begin-of-sequence and/or Final Tokens

Kallini et al. (2024) and Luo et al. (2024a) omitted a begin-of-sequence (BOS) token during tokenization when training and testing models on forward and reversed text. This oversight, though seemingly minor, makes the perplexity of forward and backward factorizations of the same sequence incomparable (Pfau et al., 2023). For a sequence of tokens $X_1, X_2, \ldots, X_n$, without a BOS token, the forward factorization computes the joint probability as $P(X_2|X_1)P(X_3|X_1, X_2) \ldots P(X_n|X_1, \ldots, X_{n-1})$, missing $P(X_1)$, while the backward factorization computes $P(X_{n-1}|X_n)P(X_{n-2}|X_{n-1}, X_n) \ldots P(X_1|X_2, \ldots, X_n)$, missing $P(X_n)$. This results in inequivalent factorizations of $P(X_1, X_2, \ldots, X_n)$. Including a BOS token ensures consistent factorizations, with $P(X_1|\text{BOS})$ and $P(X_n|\text{BOS})$, aligning the forward and backward probabilities as equivalent, as supported by the proof. In Section 4 (RQ3) of Yu et al. (2025), when comparing step-wise loss differences between forward and reversed-trained models, the first and last tokens of text sequences are excluded. For a sequence $X_1, X_2, \ldots, X_n$, this corresponds to comparing the probabilities: $P(X_2|X_1)P(X_3|X_1, X_2) \ldots P(X_{n-1}|X_1, \ldots, X_{n-2})$ and $P(X_{n-1}|X_n)P(X_{n-2}|X_n, X_{n-1}) \ldots P(X_2|X_n, \ldots, X_3)$, which is inconsistent with the theoretical comparison.

## A.2 Retraining Tokenizer on Backward Text

Both Luo et al. (2024a) and Papadopoulos et al. (2024) explored retraining tokenizers on reversed text. While Papadopoulos et al. (2024) did so for thoroughness and reported limited impact compared to reusing the original tokenizer, this approach is conceptually flawed for precisely testing theoretical forward-backward equivalence. As Luo et al. (2024a) demonstrated, retraining a Byte Pair Encoding (BPE) tokenizer (Radford et al., 2019; Gage, 1994) on character-reversed text necessarily generates a different vocabulary and tokenization compared to the original forward text. Consequently, the forward and backward passes effectively process distinct sequences, negating the theoretical foundation for comparing them.

## A.3 Mixing Logical Reversal with Token Reversal

Papadopoulos et al. (2024) and Zhang et al. (2025) investigated theoretical reasons for the observed discrepancy between forward and backward model perplexities, a deviation from theoretical predictions. Both employed controlled experiments targeting computational asymmetry, such as Zhang et al. (2025)'s analysis of multiplication (p×q=m vs. m=p×q). However, these experimental setups are flawed because they model logical reversal rather than the strict token sequence reversal required by the underlying theoretical proof. Papadopoulos et al. (2024)'s prime factorization study suffered from the same issue. Since the forward and backward tasks in these experiments do not operate on true token reversals, they fail to meet the conditions for theoretical equivalence, thereby invalidating the authors' explanations for the perplexity asymmetry observed in language models.

# B Proof of Perplexity Equivalence in Sequences of Forward and Backward Orderings

The general result (Sec. 2) encompasses perplexities of forward and backward factorized sequences as special cases. Forward perplexity ($\text{PP}_\text{fwd}$), corresponds to the natural token order, $\sigma(i) = i$ with definition

$$\text{PP}_\text{fwd} = \exp\left(-\frac{1}{n}\sum_{i=1}^{n}\ln P(X_i|X_0, X_1, \ldots, X_{i-1})\right).$$

As a direct instance of $\text{PP}_\sigma$ for the identity permutation, $\text{PP}_\text{fwd}$ immediately simplifies to the form in Equation equation 1

$$\text{PP}_\text{fwd} = \exp\left(-\frac{1}{n}\ln P(X_0, X_1, \ldots, X_n)\right).$$

Similarly, backward perplexity ($\text{PP}_{\text{bwd}}$) corresponds to the permutation $\sigma(i) = n - i + 1$, processing tokens in reverse order $(X_n, X_{n-1}, \ldots, X_1)$, with each token conditioned on $X_0$ and subsequent tokens:

$$\text{PP}_{\text{bwd}} = \exp\left(-\frac{1}{n} \sum_{i=1}^{n} \ln P(X_{\sigma(i)} | X_0, X_{\sigma(1)}, \ldots, X_{\sigma(i-1)})\right),$$

where $X_{\sigma(i)} = X_{n-i+1}$. This also simplifies to

$$\text{PP}_{\text{bwd}} = \exp\left(-\frac{1}{n} \ln P(X_0, X_1, \ldots, X_n)\right).$$

The proof shows the theoretical equivalence of perplexity regardless of the factorization order chosen for the sequence, provided the underlying probability model is consistent and adheres to the chain rule of probability.

## C  Intuitive Example with Three Variables

To make the generalization to arbitrary orderings more intuitive, consider a joint probability distribution over three random variables $A$, $B$, and $C$. The joint probability $P(A, B, C)$ can be factorized in several equivalent ways using the chain rule:

**Forward Order (A, B, C)**

$$P(A, B, C) = P(A) \cdot P(B|A) \cdot P(C|A, B)$$

**Backward Order (C, B, A)**

$$P(A, B, C) = P(C) \cdot P(B|C) \cdot P(A|B, C)$$

**Other Permutations**

$$P(A, B, C) = P(B) \cdot P(A|B) \cdot P(C|A, B)$$
$$P(A, B, C) = P(B) \cdot P(C|B) \cdot P(A|B, C)$$
$$P(A, B, C) = P(C) \cdot P(A|C) \cdot P(B|A, C)$$
$$P(A, B, C) = P(A) \cdot P(C|A) \cdot P(B|A, C)$$

These factorizations correspond to the six possible orderings of three variables: $(A, B, C)$, $(A, C, B)$, $(B, A, C)$, $(B, C, A)$, $(C, A, B)$, and $(C, B, A)$. If we calculate the perplexity using any of these orderings, we should arrive at the same value:

$$PP_\sigma = \exp\left(-\frac{1}{3} \ln P(A, B, C)\right)$$

## D  Training Setups

Building on the theoretical proof and experimental deviations we identify in prior work, we re-evaluate (Luo et al., 2024a) using theory-aligned experimental protocols. We highlight key training and evaluation details to specify the conditions required for assessing perplexity equivalence across sequence factorizations.

### D.1  Tokenization Direction

We adopt GPT-2's tokenization strategy (Radford et al., 2019), based on Byte Pair Encoding (BPE) (Gage, 1994) for word segmentation (Sennrich et al., 2016). The tokenizer, trained on neuroscience publications (1.3 billion tokens) in the forward direction with a vocabulary of 50,257 tokens, is used for all models. Importantly, the same tokenizer is used for both backward and permuted models, contrasting prior work (Luo et al., 2024a). For backward models, forward text is tokenized into token IDs and reversed within the context window. For permuted models, the tokenized forward text is rearranged in a fixed order within the context window.

### D.2 Special Tokens

To align with the proof of perplexity equivalence, all training and validation sequences fully span the GPT-2 context window (1,024 tokens), each prefixed with a begin-of-sequence token (Papadopoulos et al., 2024), excluding document separators (unlike the original GPT-2; Radford et al. 2019 and padding tokens). This ensures the sequence of different factorizations whose joint probabilities are comparable, as the first real token is not excluded from the joint probability factorization due to the shifted-by-one calculation in next-token prediction loss, maintaining consistency across factorizations, unlike previous work (Pfau et al., 2023). In addition, during loss computation, the starting token's probability is masked out in the softmax operation to align closely with the proof.

### D.3 Model Variants

We train GPT-2 models of varying sizes using three data orders: forward, backward, and permuted. All models are optimized with standard autoregressive loss. To ensure consistency, all models are trained on identical data sequences with all randomness sources fixed, differing only in the arrangement of data within the context window or the random seed.

### D.4 Training Data

We trained GPT-2 variants from scratch using data collected by Luo et al. (2024b), comprising 1.3 billion tokens from neuroscience publications (abstracts and full articles) spanning 2002–2022. The dataset was split randomly, with 90% allocated for training and 10% for validation. To align with the proof of perplexity equivalence, all training and validation sequences fully span the GPT-2 context window (1,024 tokens), each prefixed with a begin-of-sequence token, excluding document separators and padding tokens. The last sequences of training and validation sets shorter than 1,023 tokens were discarded, yielding 995,270 training sequences and 9,413 validation sequences.

### D.5 Training details

We trained variants of GPT-2 models using Huggingface implementations. We used a batch size of 16 for GPT-2 124M (8 for GPT-2 355M and 4 for GPT-2 774M) and a chunk size of 1024. Training involved the use of the AdamW optimizer (Loshchilov & Hutter, 2019) with a learning rate of 2e-5 and a cosine learning rate scheduler (i.e., learning rate decays following a cosine schedule over training epochs). We applied gradient accumulation steps set at 8. Five training epochs were performed, along with a warm-up step of 0.03 and a weight decay rate of 0.001. bf16 mixed precision training and data parallelism were employed. We used 4 Nvidia A100 (80GB) GPUs hosted on Microsoft Azure.

## E Evaluation Setups

### E.1 BrainBench

BrainBench (Luo et al., 2024b) is a benchmark consists of 200 test cases from abstracts in the *Journal of Neuroscience* published in 2023. These abstracts are categorized into five sections: Behavioral/Cognitive, Systems/Circuits, Neurobiology of Disease, Development/Plasticity/Repair, and Cellular/Molecular.

Each test case contains a published abstract and an altered version crafted by neuroscientists (see details in Luo et al. (2024b)). These modifications, though minimal, significantly change the results—for instance, by changing the roles of brain regions or reversing a result's direction (e.g., from "decreases" to "increases"). The altered abstracts remain logically coherent despite the changes.

The BrainBench task is to identify the correct study outcome by choosing between the original abstract and its altered counterpart.

## E.2   BrainBench Model Evaluation

Two versions of the abstracts from each test case were presented to models separately. We measured the perplexity of both passages and used perplexity as the indicator of whether models favor one abstract or the other.

Perplexity measures the degree of uncertainty of a model when generating a particular sequence of text and is defined as the exponentiated average negative log-likelihood of a tokenized sequence. If we have a tokenized abstract sequence $X = (X_0, X_1, \ldots, X_t)$, then the perplexity of $X$, given a model parameterized by $\theta$ is,

$$PP(X) = \exp\left\{ -\frac{1}{t} \sum_{i}^{t} \ln p_\theta(X_i | X_{<i}) \right\} \tag{2}$$

where $\ln p_\theta(X_i | X_{<i})$ is the log-likelihood of the $i$th token conditioned on the preceding tokens $X_{<i}$ according to the model. Given both the original and the altered abstracts, we used the abstract with lower perplexity as the model's decision and evaluated the overall accuracy across the entire BrainBench dataset accordingly.

## E.3   BrainBench Human Evaluation

Previous work (Luo et al., 2024b) collected human judgements from 171 neuroscience experts on BrainBench. These data are publicly available[1] and provide a useful comparison to LLM performance.

## E.4   Statistical Testing

To test the effects of model size and training direction on prediction accuracy, we conducted a repeated-measures Analysis of Variance (ANOVA). The dependent variable was prediction correctness for each Brain-Bench item. Model size and direction were included as fixed factors, with model size coded as a continuous variable and direction binary-coded as a categorical variable. Model was treated as a within-subjects factor to account for repeated measurements. The analysis was implemented using the `aov()` function in R, with the Error term specified to accommodate the repeated-measures design. Both main effects and the interaction between model size and direction were examined.

## F   Additional Results

Table S.1: Comparison of Statistical Metrics for Fwd and Bwd Directions Across GPT-2 Models Across Three Initializations. Significance denoted by *** indicates p < 0.001.

| Model | Direction | Pearson $r$ | $t$-stat | Cohen's $d$ |
|-------|-----------|-------------|----------|-------------|
| 124M  | Fwd       | $0.999 \pm 0.000$*** | $30.185 \pm 11.349$*** | $0.311 \pm 0.117$ |
|       | Bwd       | $0.999 \pm 0.000$*** | $10.930 \pm 28.250$*** | $0.113 \pm 0.291$ |
| 355M  | Fwd       | $0.999 \pm 0.000$*** | $-3.561 \pm 1.476$*** | $0.037 \pm 0.015$ |
|       | Bwd       | $0.999 \pm 0.000$*** | $1.597 \pm 3.837$*** | $0.016 \pm 0.040$ |
| 774M  | Fwd       | $0.999 \pm 0.000$*** | $-13.640 \pm 6.063$*** | $0.141 \pm 0.062$ |
|       | Bwd       | $0.999 \pm 0.000$*** | $-9.885 \pm 3.716$*** | $0.102 \pm 0.038$ |

---

[1]https://github.com/braingpt-lovelab/BrainBench

Table S.2: Comparison of Statistical Metrics for Perm Direction Across GPT-2 Models Across Three Initializations. Significance denoted by *** indicates $p < 0.001$.

| Model | Direction | Pearson $r$ | $t$-stat | Cohen's $d$ |
|-------|-----------|-------------|----------|-------------|
| 124M | Perm | $0.995 \pm 0.000$*** | $106.154 \pm 83.993$*** | $1.094 \pm 0.866$ |
| 355M | Perm | $0.995 \pm 0.000$*** | $44.731 \pm 23.499$*** | $0.461 \pm 0.242$ |
| 774M | Perm | $0.994 \pm 0.000$*** | $30.051 \pm 17.785$*** | $0.310 \pm 0.183$ |

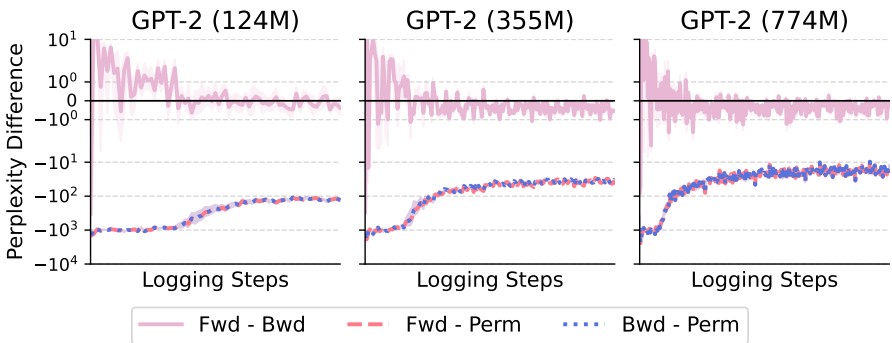

Figure S.1: **Training perplexity differences across model sizes and training directions.**

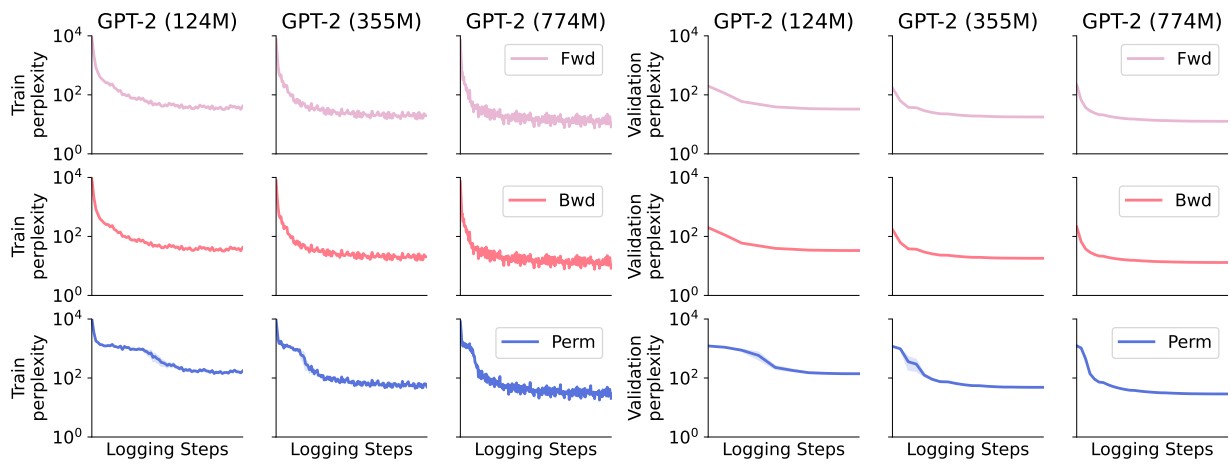

Figure S.2: **Training and Validation perplexities across model sizes and training directions.**

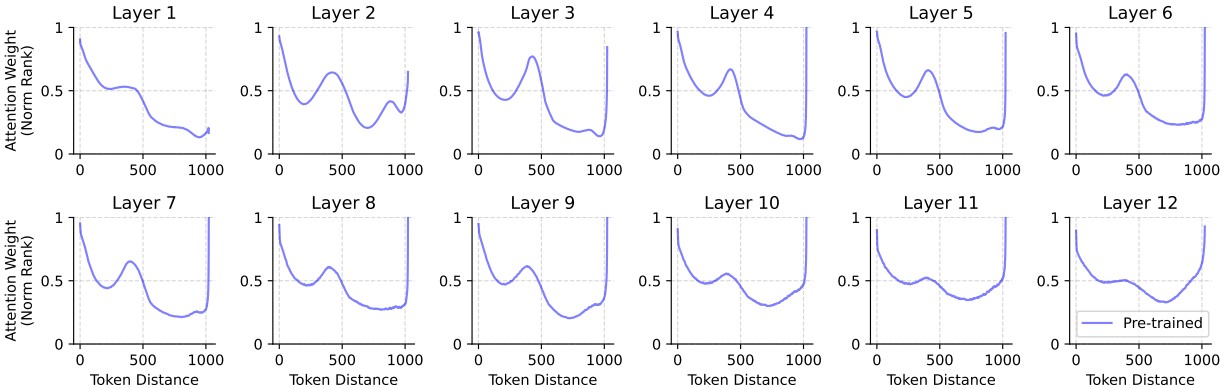

Figure S.3: **Self-attention is biased toward adjacent and long-range tokens (GPT-2 124M Pre-trained).** Normalized attention rank (0–1) is plotted as a function of token distance within the context, averaged across heads, sampled sequences, and layers. Sequences are all 1,024 tokens with BOS prepended, sampled from the first 10K entries from the Pile (Gao et al., 2020).

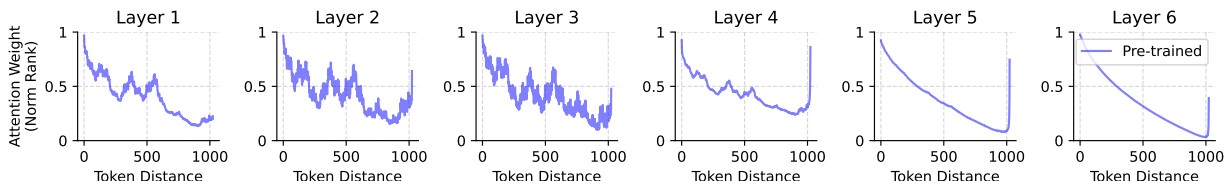

Figure S.4: **Self-attention is biased toward adjacent and long-range tokens (Pythia 70M Pre-trained).** Normalized attention rank (0–1) is plotted as a function of token distance within the context, averaged across heads, sampled sequences, and layers. Sequences are all 1,024 tokens with BOS prepended, sampled from the first 10K entries from the Pile (Gao et al., 2020).

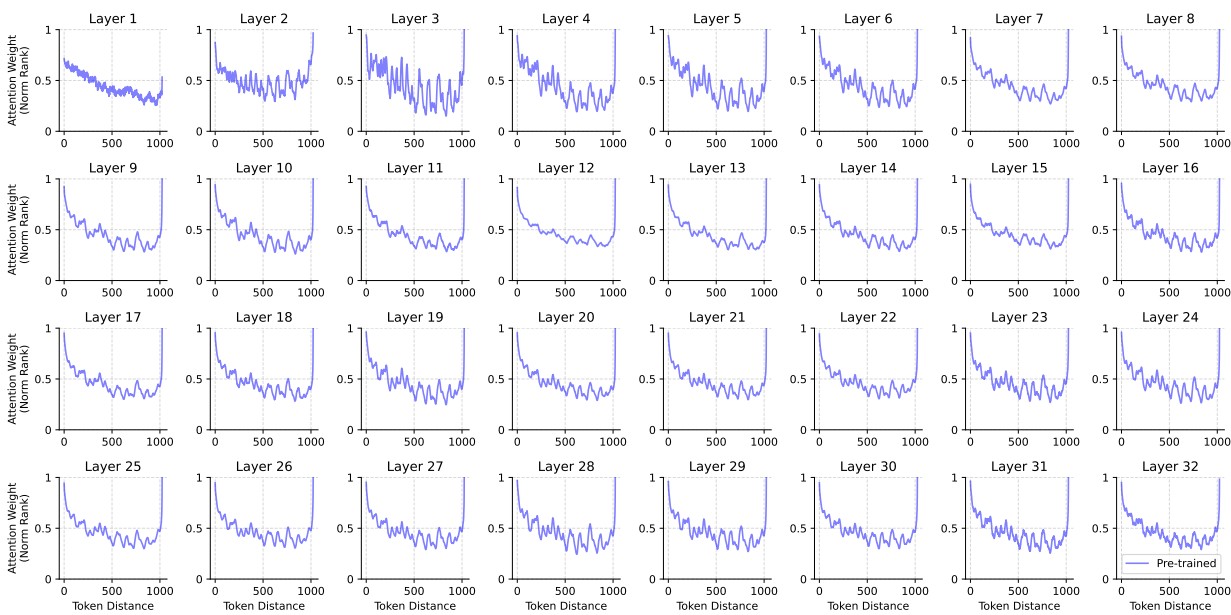

Figure S.5: **Self-attention is biased toward adjacent and long-range tokens (Llama-2 7B Pre-trained).** Normalized attention rank (0–1) is plotted as a function of token distance within the context, averaged across heads, sampled sequences, and layers. Sequences are all 1,024 tokens with BOS prepended, sampled from the first 10K entries from the Pile (Gao et al., 2020).

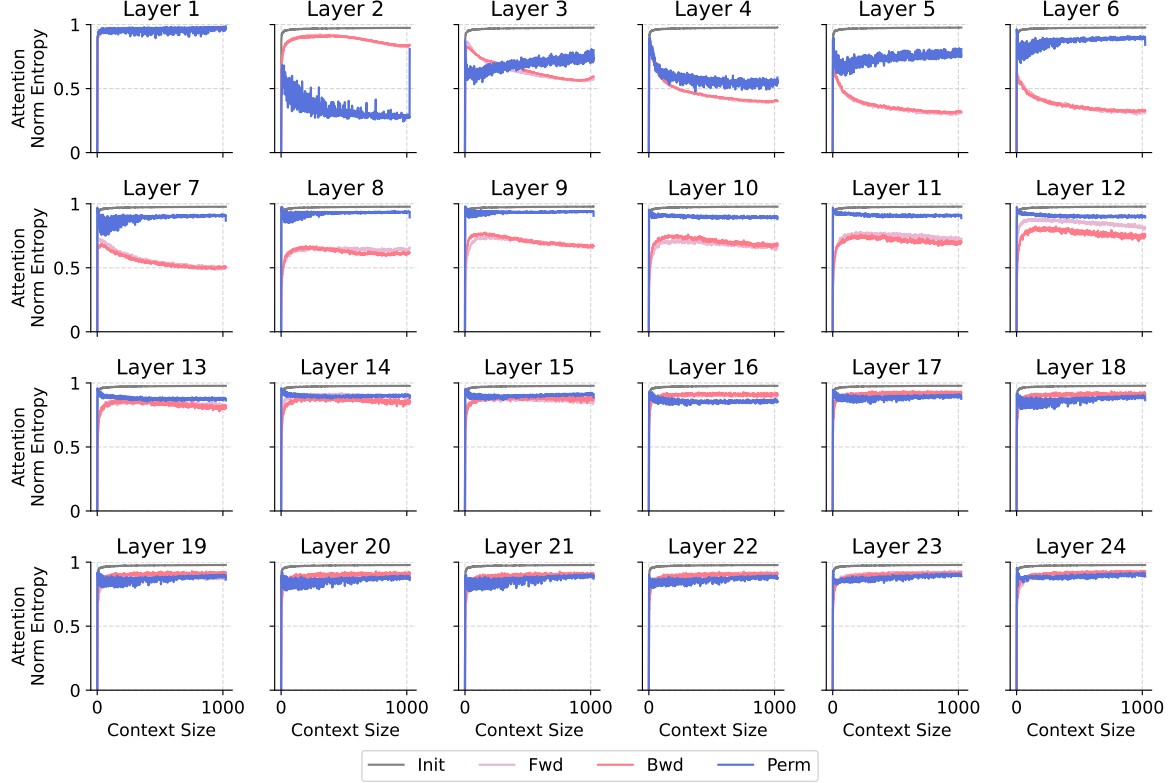

Figure S.6: **Attention entropy across three data orders (GPT-2 355M, seed1).**

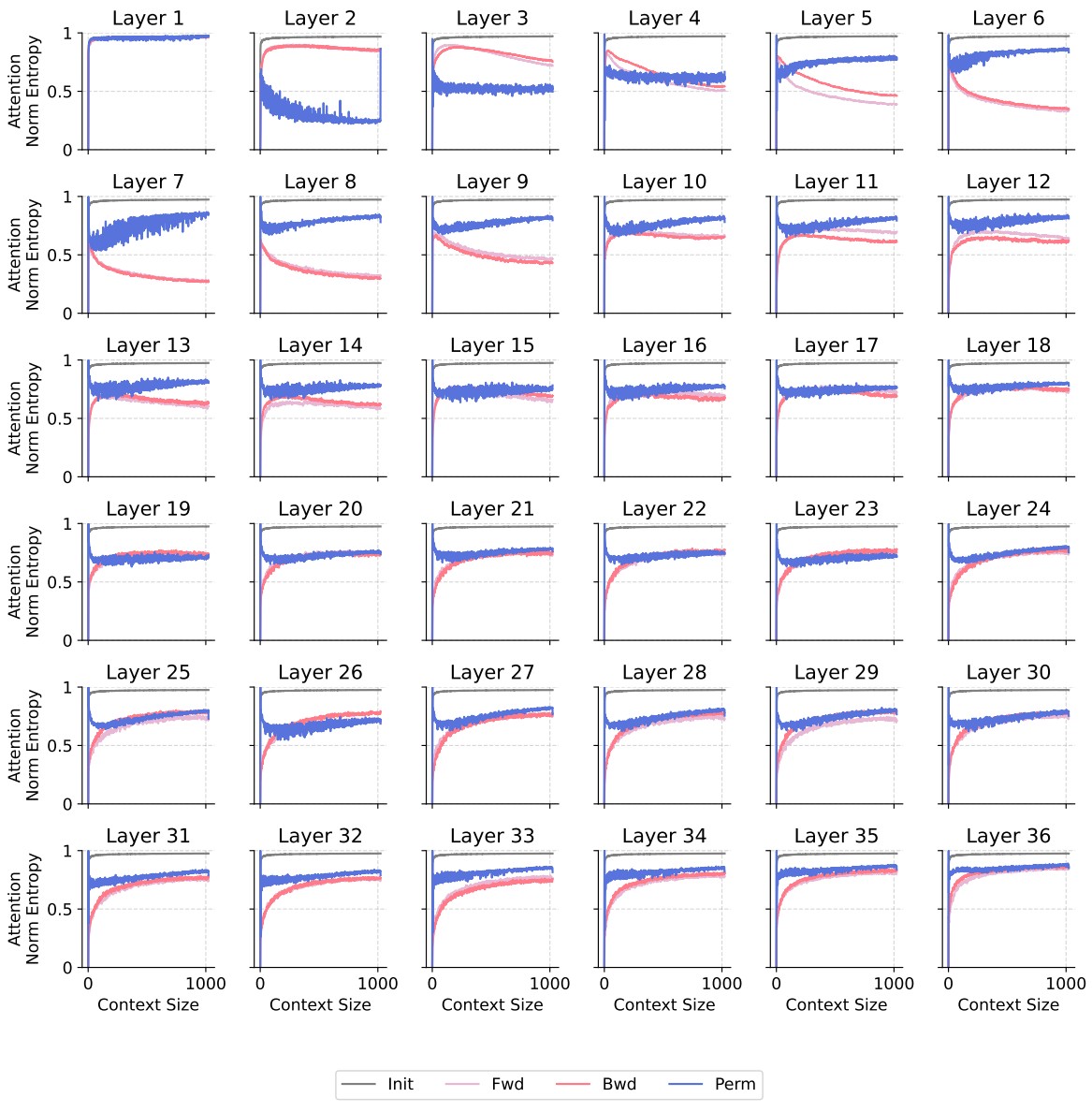

Figure S.7: **Attention entropy across three data orders (GPT-2 774M, seed1).**

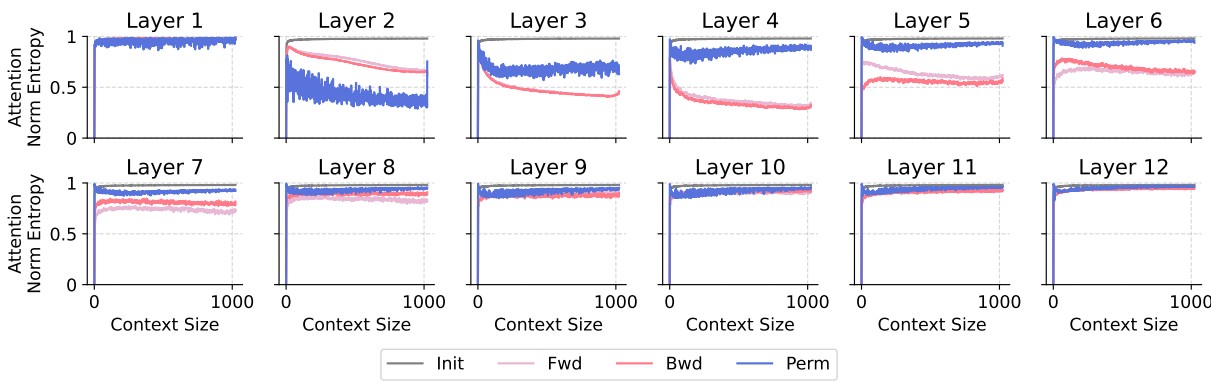

Figure S.8: **Attention entropy across three data orders (GPT-2 124M, seed2).**

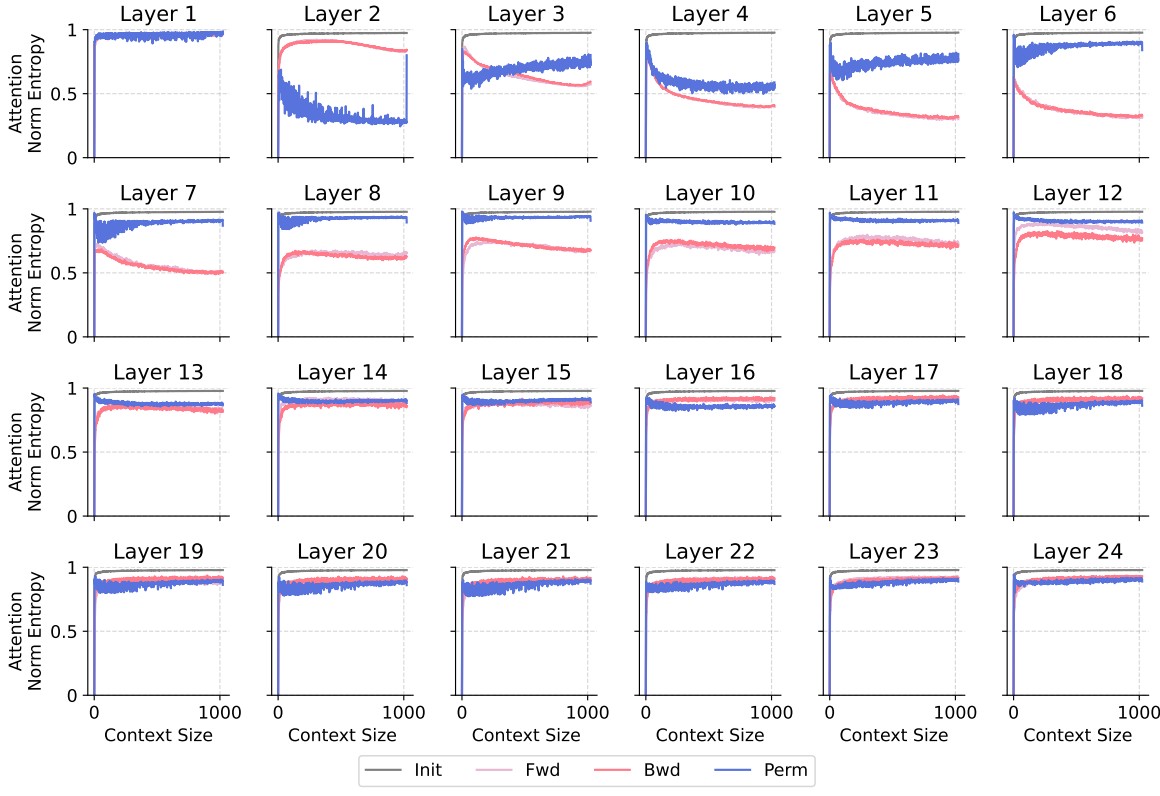

Figure S.9: **Attention entropy across three data orders (GPT-2 355M, seed2).**

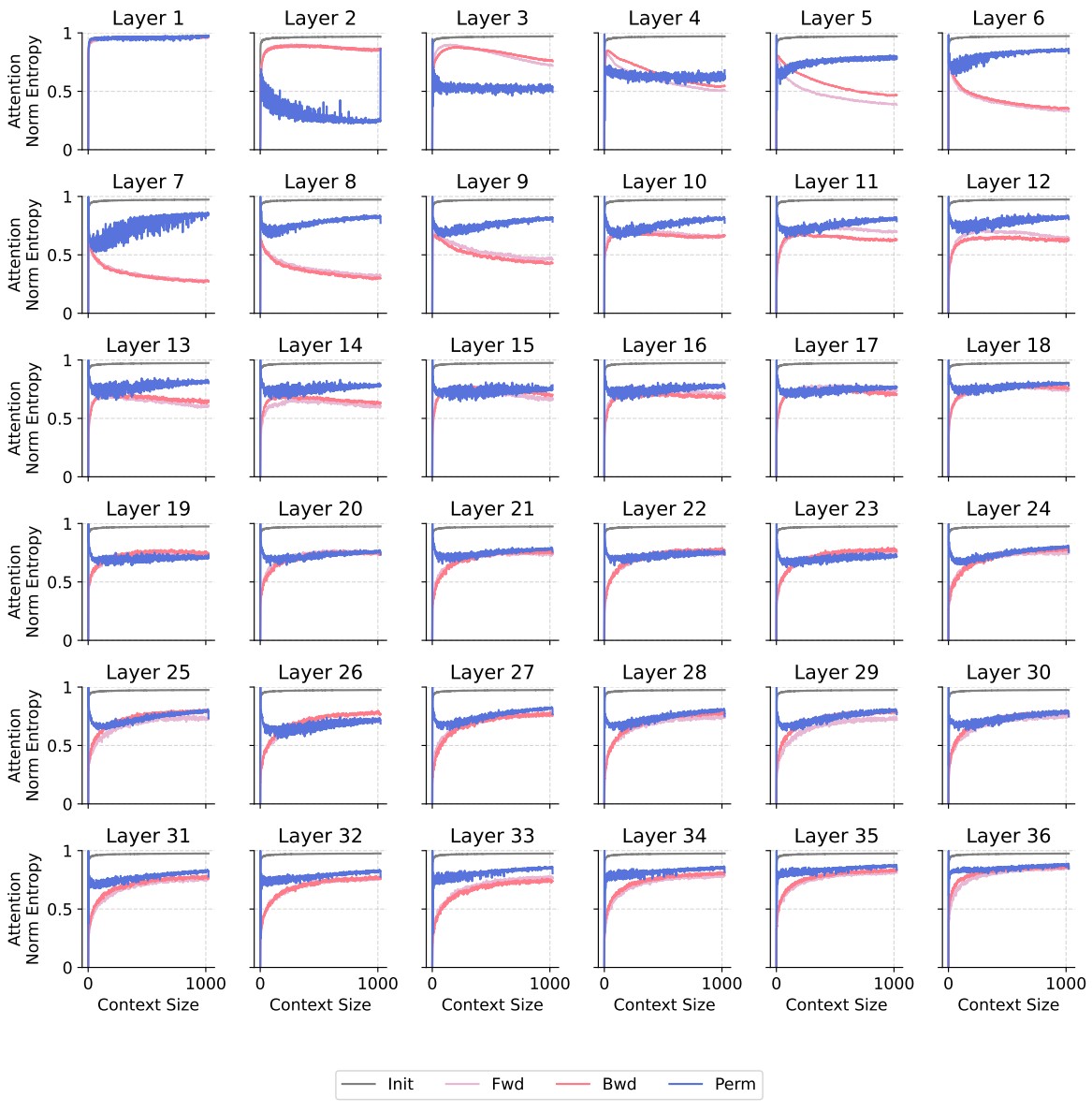

Figure S.10: **Attention entropy across three data orders (GPT-2 774M, seed2).**

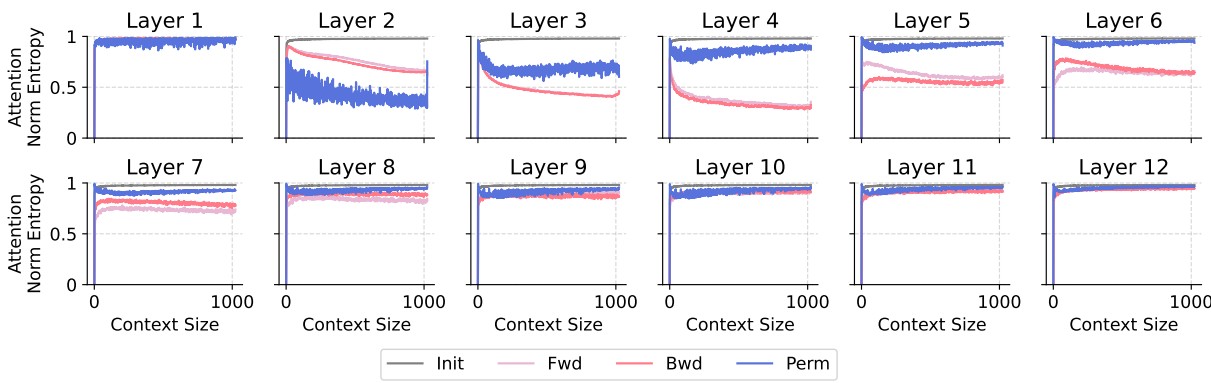

Figure S.11: **Attention entropy across three data orders (GPT-2 124M, seed3).**

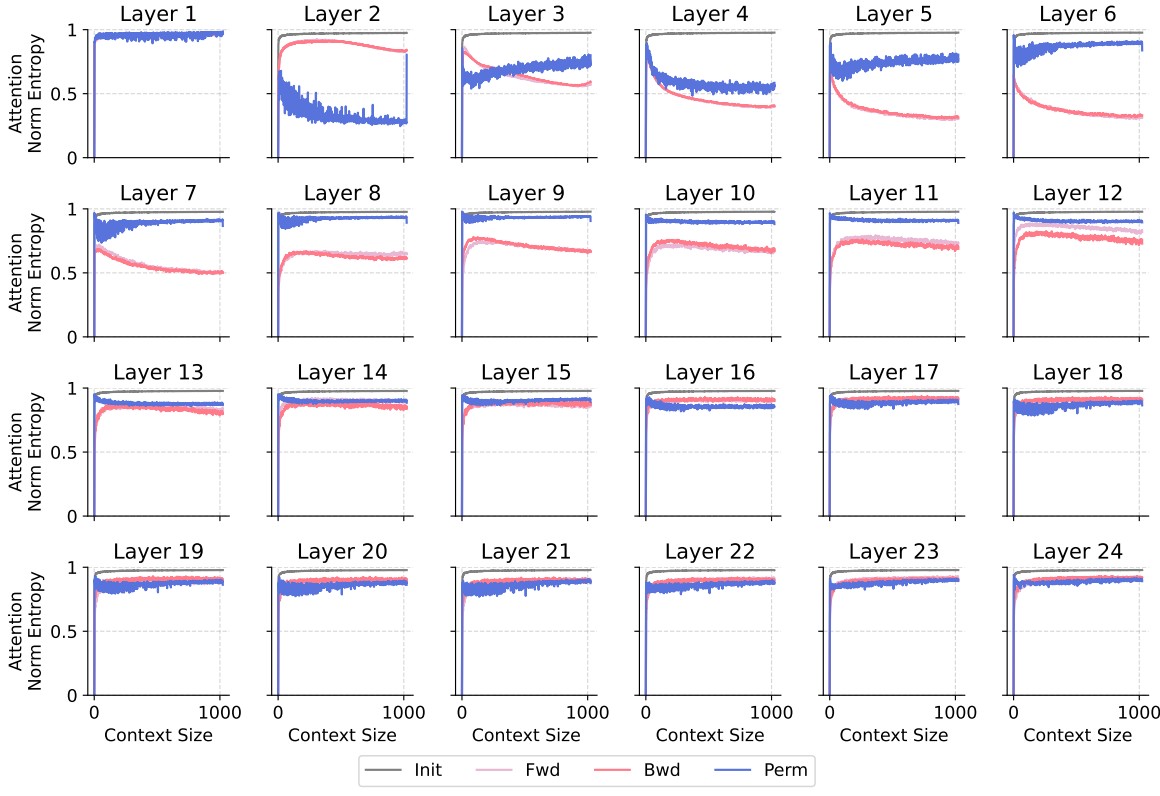

Figure S.12: **Attention entropy across three data orders (GPT-2 355M, seed3).**

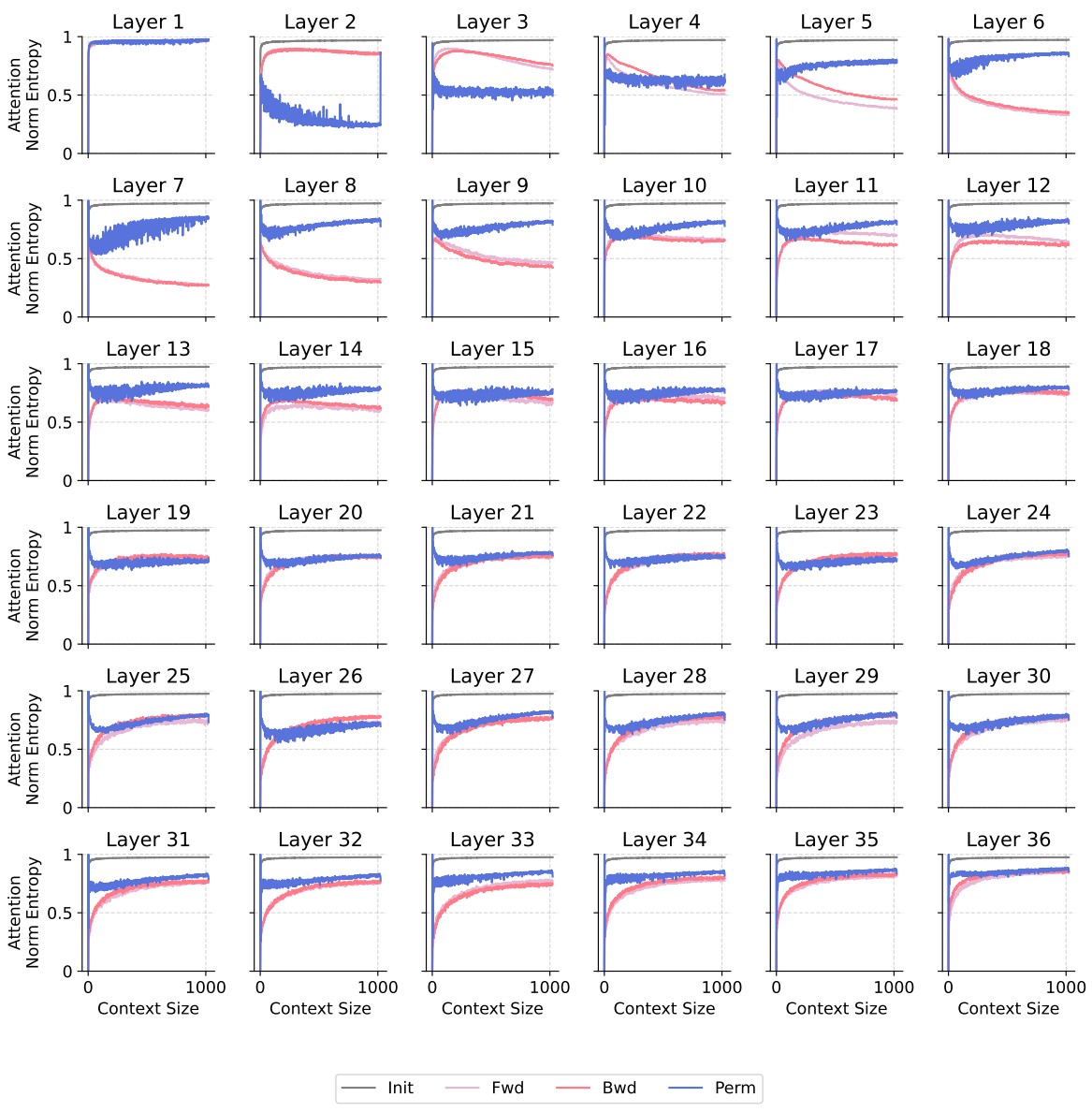

Figure S.13: **Attention entropy across three data orders (GPT-2 774M, seed3).**

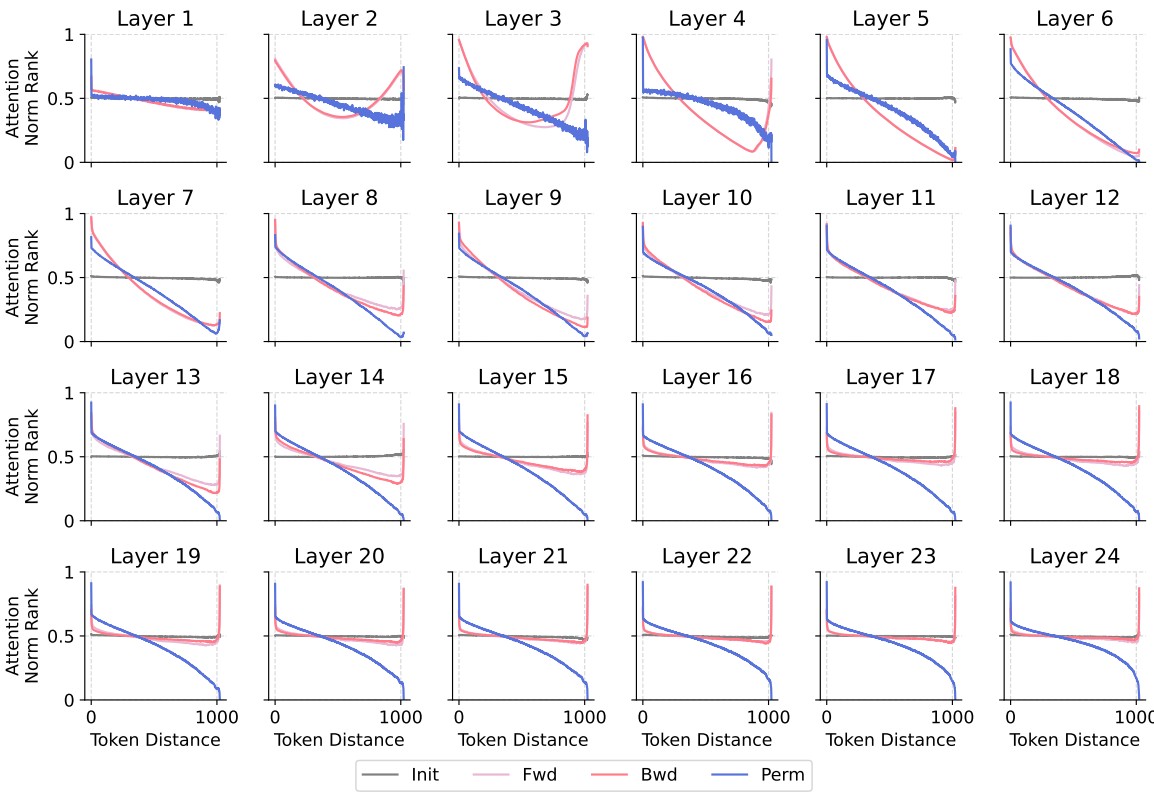

Figure S.14: **Self-attention biases (GPT-2 355M, seed1).**

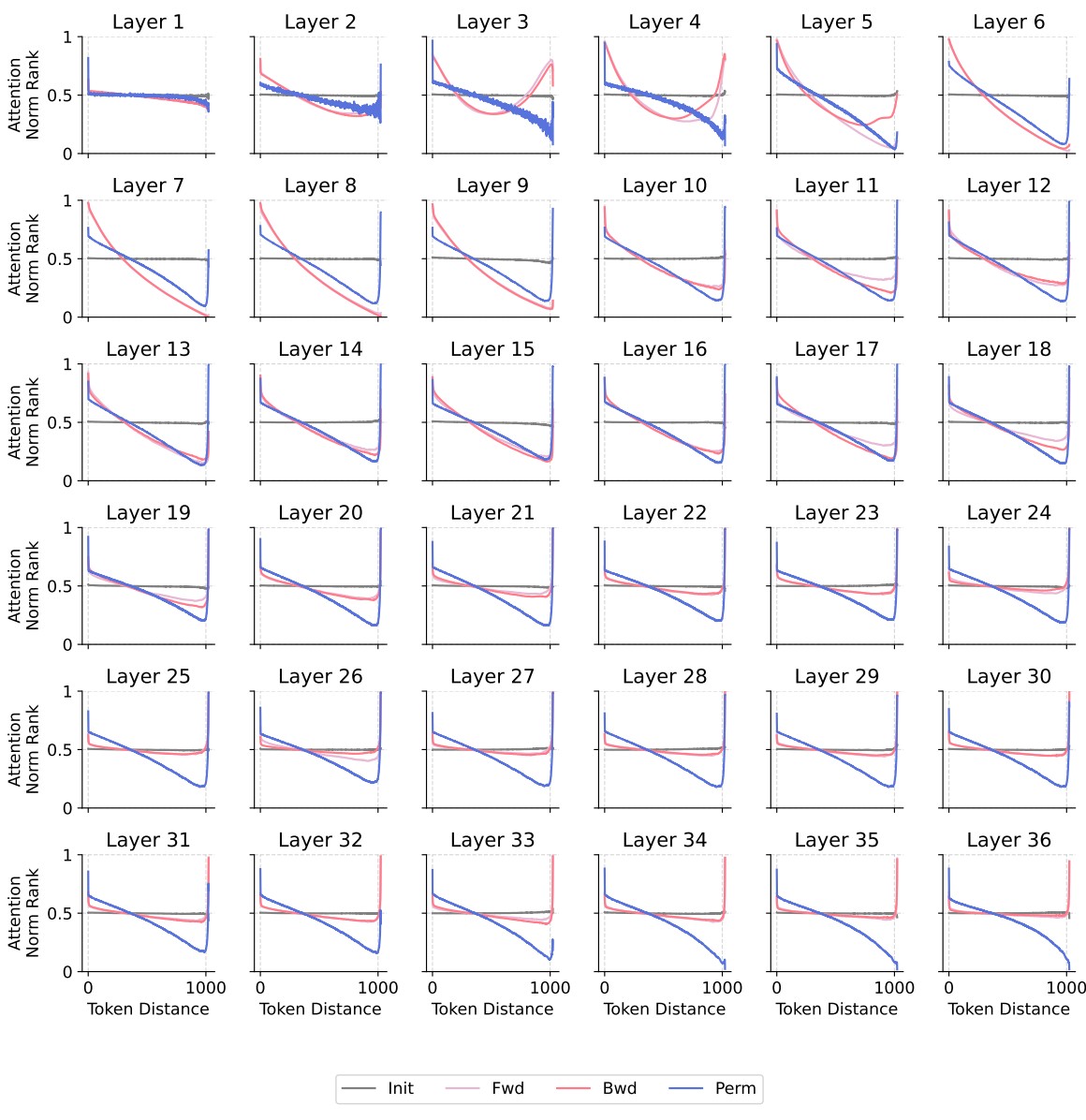

Figure S.15: **Self-attention biases (GPT-2 774M, seed1).**

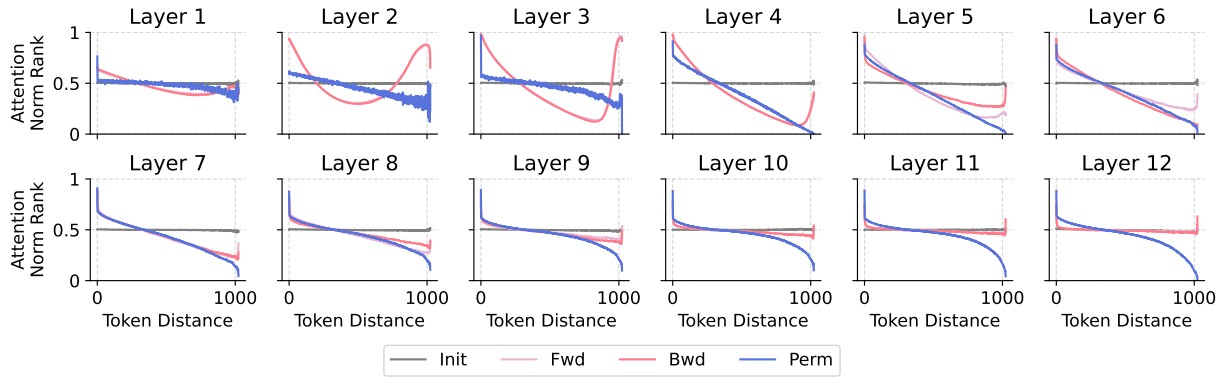

Figure S.16: **Self-attention biases (GPT-2 124M, seed2).**

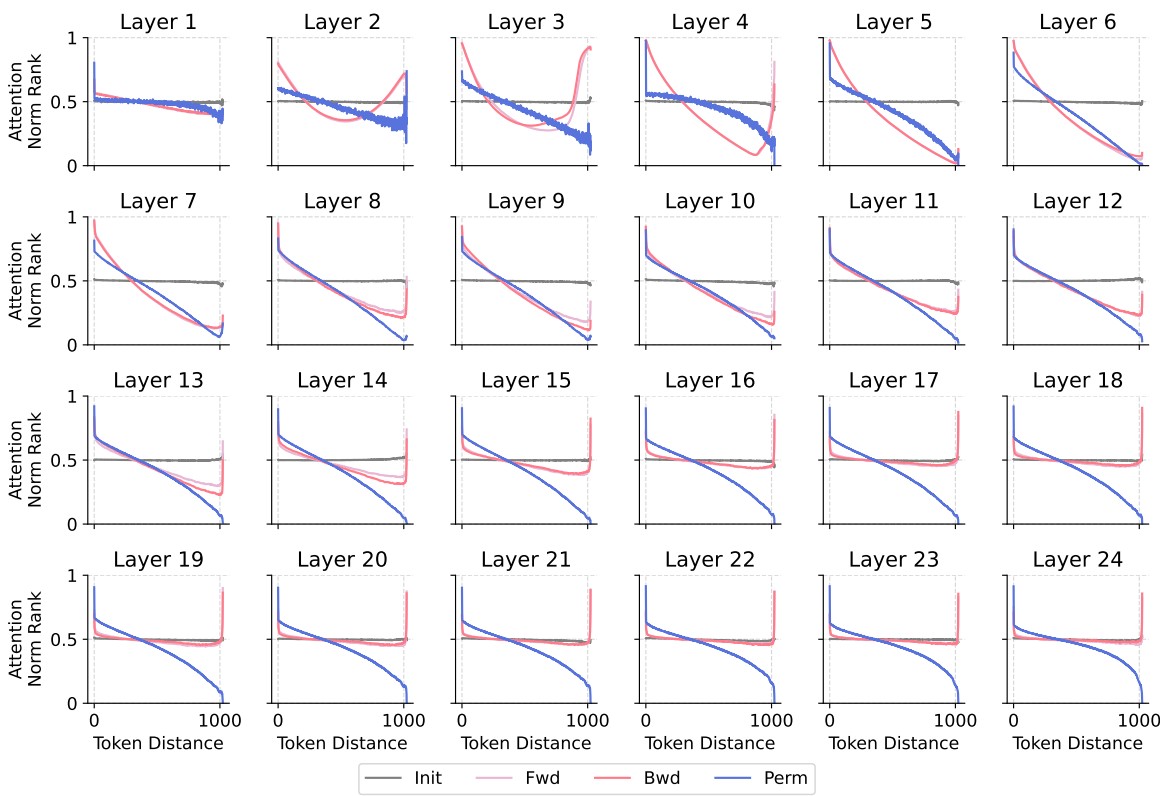

Figure S.17: **Self-attention biases (GPT-2 355M, seed2).**

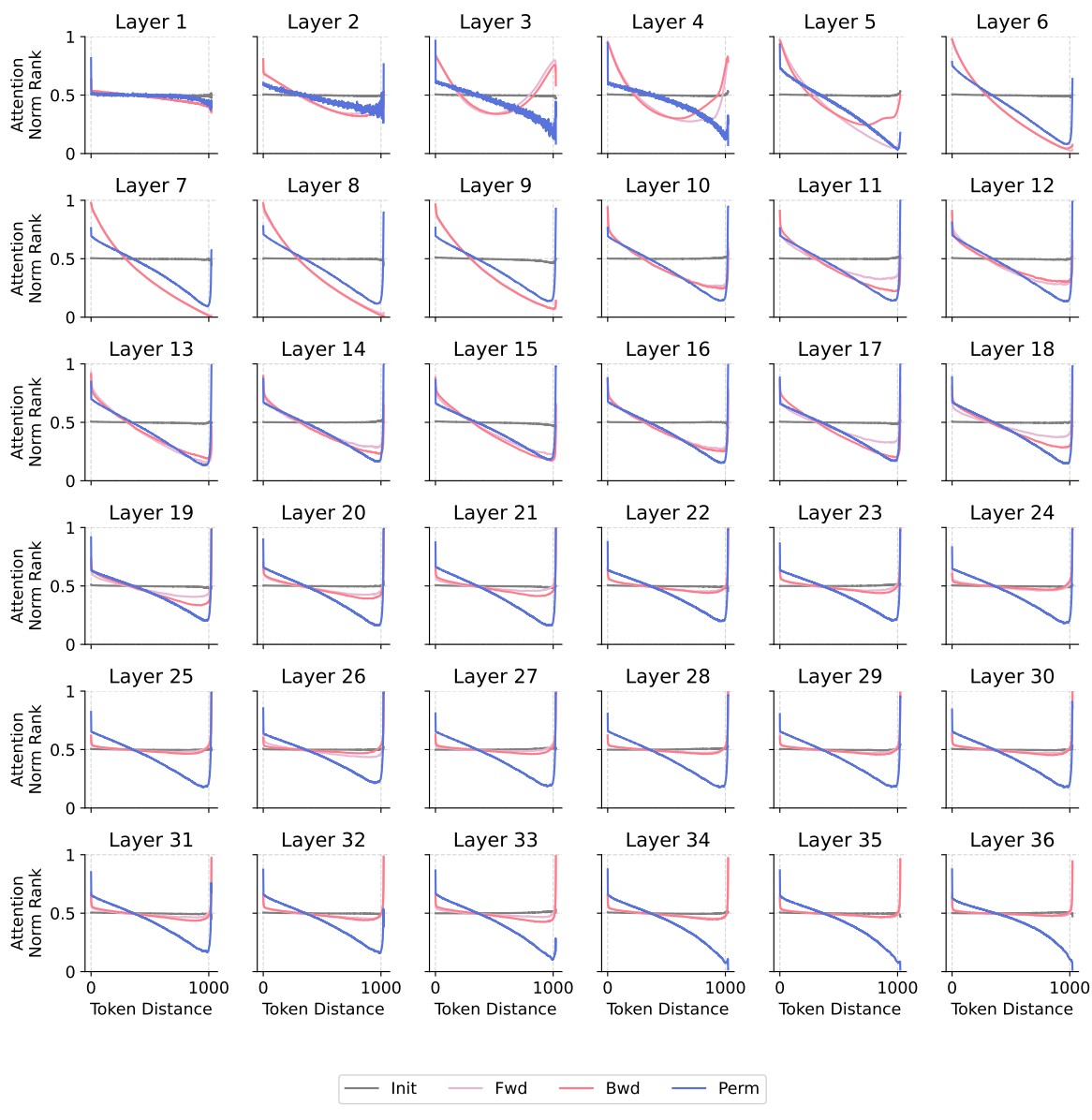

Figure S.18: **Self-attention biases (GPT-2 774M, seed2).**

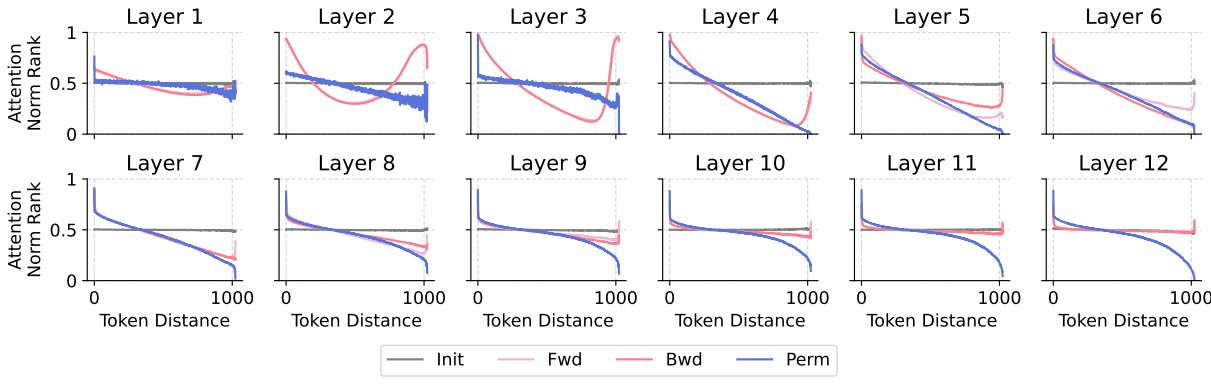

Figure S.19: **Self-attention biases (GPT-2 124M, seed3).**

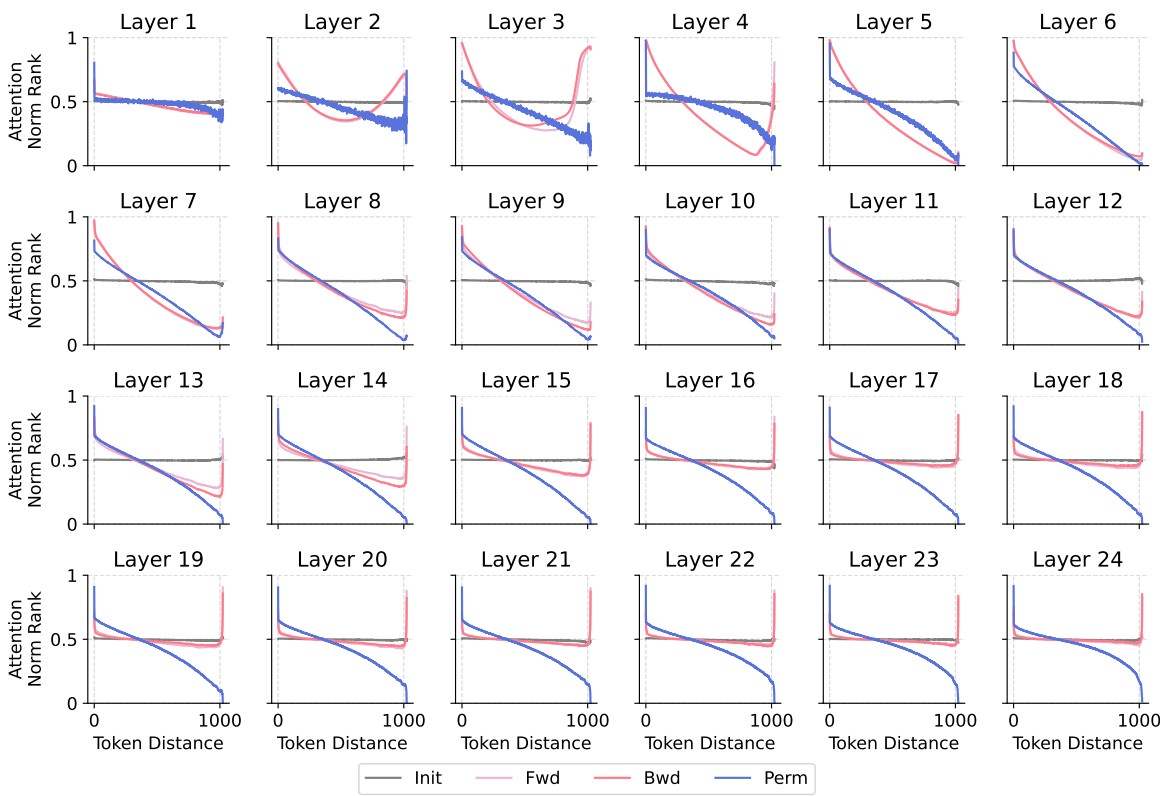

Figure S.20: **Self-attention biases (GPT-2 355M, seed3).**

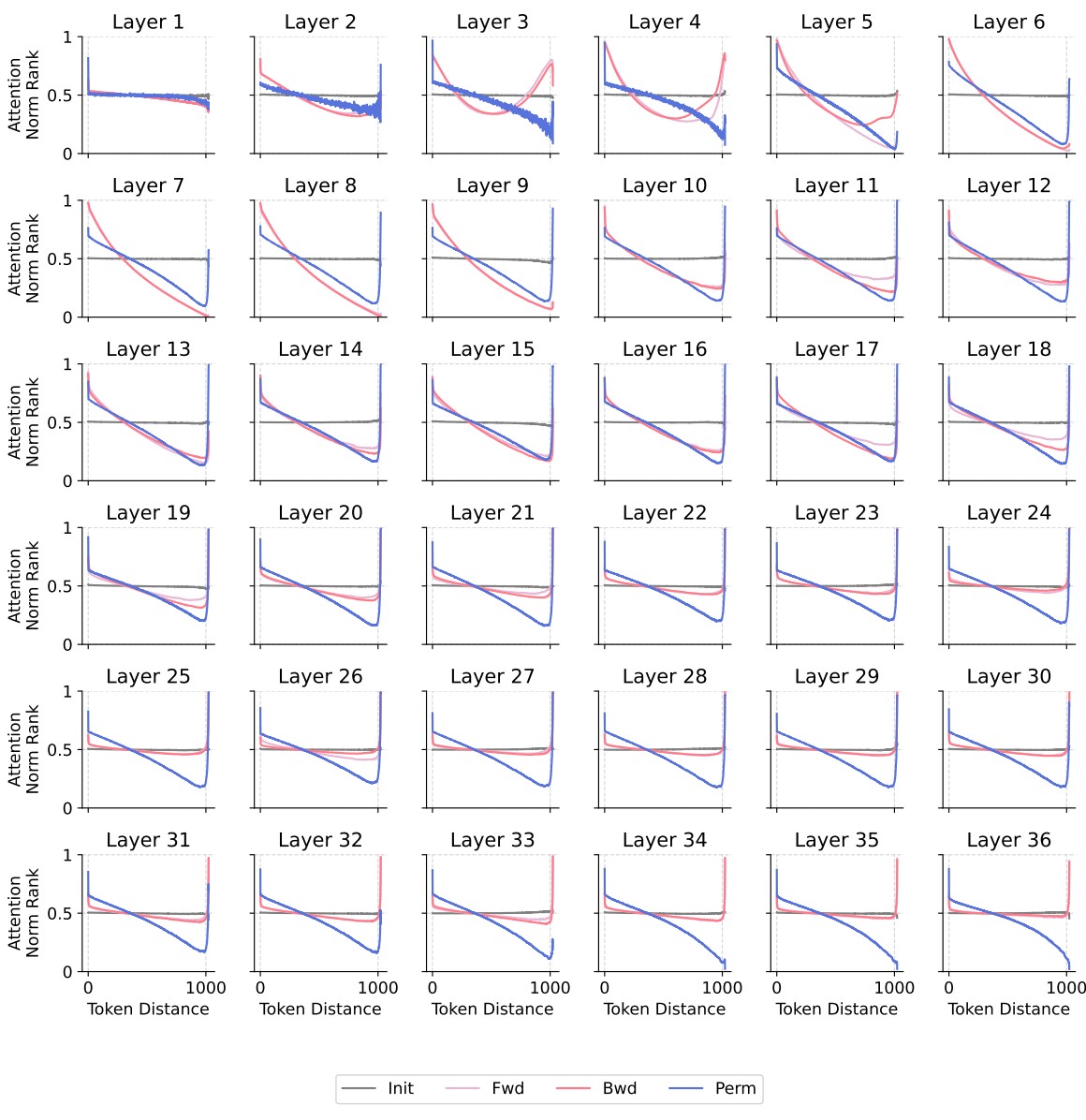

Figure S.21: **Self-attention biases (GPT-2 774M, seed3).**

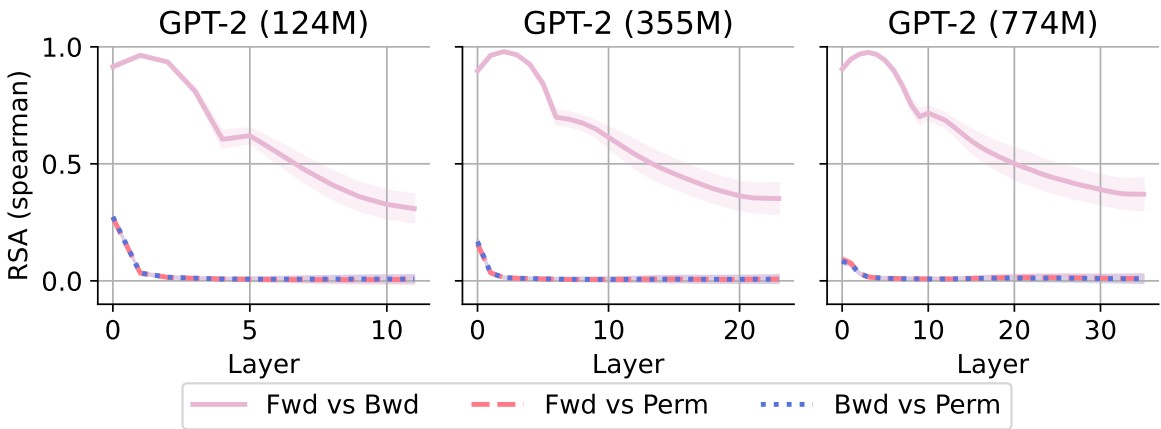

Figure S.22: **Representational similarities across model training directions (seed2)**

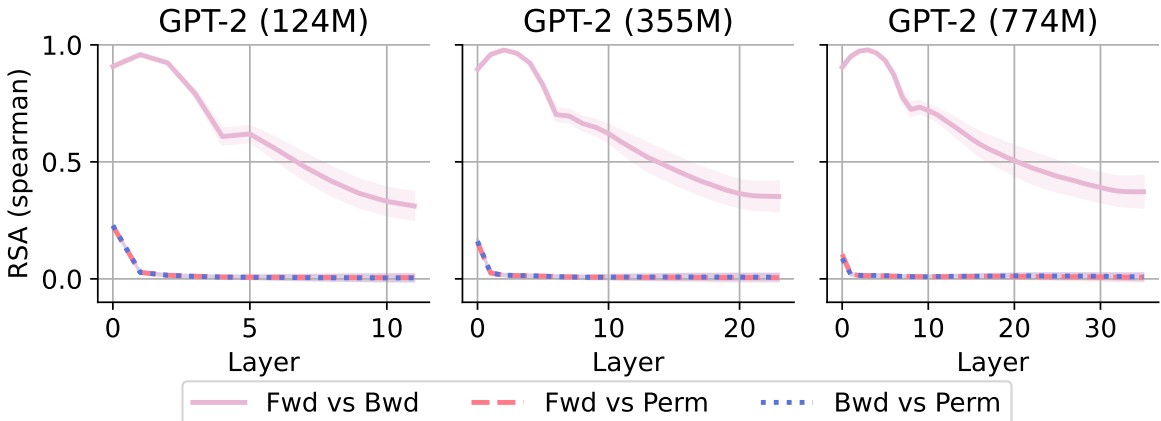

Figure S.23: **Representational similarities across model training directions (seed3)**

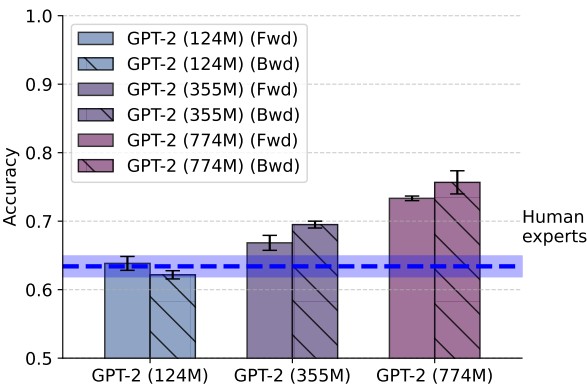

Figure S.24: **BrainBench performance of GPT-2 models trained forward and backward.** GPT-2 models, trained from scratch on two decades of neuroscience literature, rival or exceed human expert performance, demarcated by the blue dashed line. Models trained on the same data reversed at the token level performed similarly to their forward-trained counterparts. Error bars are standard errors of the mean.

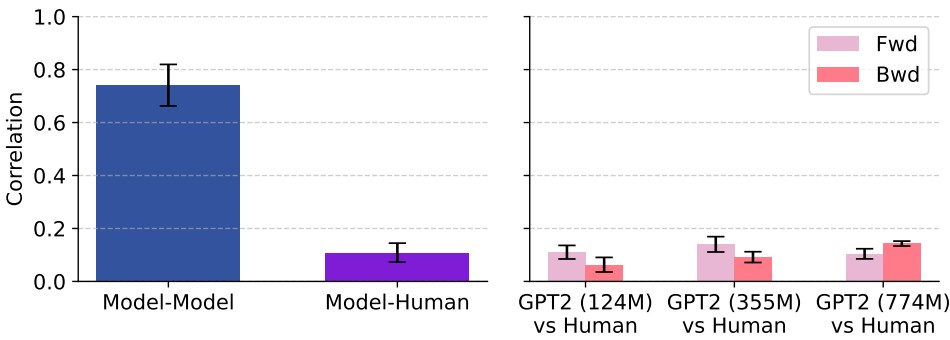

Figure S.25: **Comparison of model and human judgments on BrainBench difficulty.** Model judgments (both forward and backward-trained) correlate more strongly with each other than with human expert judgments. Backward-trained models show similar correlation to human judgments compared to forward-trained models. Error bars are one standard deviations of the mean.

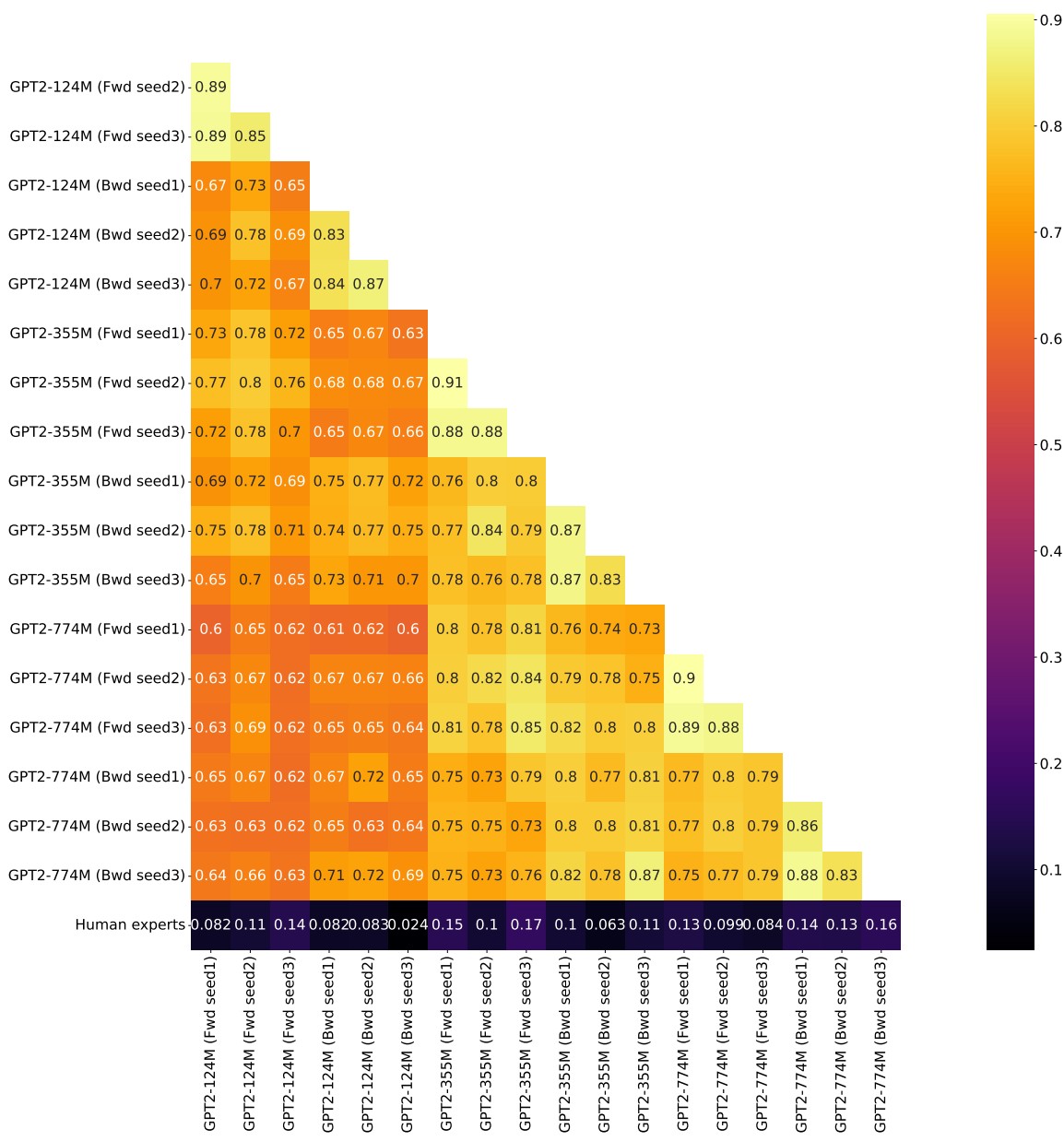

Figure S.26: **Comparison of model and human judgments on BrainBench difficulty.** Model judgments (both forward and backward-trained) correlate more strongly with each other than with human expert judgments. Backward-trained models show similar correlation to human judgments compared to forward-trained models.

