# OpenReview forum: "Probability Consistency in Large Language Models: Theoretical Foundations Meet Empirical Discrepancies"
_TMLR — Rejected by TMLR_

### Review · Reviewer_USZ8 · 2025-09-16

**Summary Of Contributions:**

This submission shows that, given a probability distribution over a finite set of sequences, permuting the sequence order should in theory keep perplexity the same. The authors also examine whether this is true in practice or not for trained models based on GPT-2. They observe that using sequences in forward and backward order often leads to similar perplexity, but perplexity is nevertheless often slightly lower for forward models. Perplexity of permuted sequences is often much higher. The authors present additional metrics/visualizations to better understand the observed discrepancies, and further measure downstream performance on the neuroscience BrainBench dataset.

The authors also claim that previous papers had critical errors in their experimental setup, in particular missing a beginning-of-sentence token (or final tokens), using different tokenizers. or using logical reversal instead of token reversal.

**Audience:**

Yes

**Audience Explanation:**

Understanding large language models and the impact of sequence ordering can be of interest to the TMLR audience. Furthermore, the results are fairly detailed.

The paper could potentially be even more interesting by considering variable length sequences, which are more realistic for large language models. For such models, if there is a fixed context window (with padding), the impact of unterminated sequences could also be examined.

Although I believe some individuals may be interested in the paper, I found the main claim about perplexity equivalence under permutations to be fairly unsurprising.

**Claims And Evidence:**

No

**Claims Explanation:**

The claims about other papers containing critical methodological flaws may be true, but the paper does not provide sufficient details to clearly establish this. For example, using a different tokenizer for reversed sentences may be a reasonable practical choice, even if probabilities are not directly comparable between different sequence orders.

I would also like additional evidence for the claims related to the BOS token. For example, in Luo et al. 2024a, the authors mentions that they add a common prefix to the prompt (section 4.2), which seems reasonable even if it differs from the methodology in the current submission. Briefly looking at Kallini et al., 2024, I could not establish with certainty if there was a glaring issue. More generally, although using a BOS token can simplify the implementation, the probability of the first token can also be defined as $P(X_1|\emptyset$), where $\emptyset$ is an empty sequence.

The notation in the section 2 proof could be improved. In particular, $P(X_0, X_{\sigma(1)}, \dots, X_{\sigma(n)}) = P(X_0, X_1, \dots, X_n)$ is not quite exact as the probability distributions on both sides of the equation are different and should ideally use a different symbol.

**Requested Changes:**

**Critical**

I would like the authors to discuss in more nuance the "critical methodological flaws". While some of these changes may break the equivalence property discussed in this submission, they may be reasonable empirically depending on what the authors of previous papers were trying to accomplish. If there are indeed flaws in previous studies, it's fine to point them out, but the discussion should be more detailed (even beyond the content of appendix A).

---

> ### Author Response · Authors · 2025-09-24
> **Response to reviewer comments**
>
> We agree that training different tokenizers for reversed sequences may be practical, but it breaks our theoretical basis for exploring joint probability equivalence. Different vocabularies invalidate our core study purpose, which requires formal equivalency across training and evaluation orders (Changes highlighted toward the end of Sec. 2).
>
> As stated in response to the first reviewer, autoregressive LLMs capture only one factorization and cannot be evaluated on different token orderings than training ones. We trained sibling models on different factorizations to assess their learned factorizations. Creating situations where models should theoretically have identical perplexity makes their divergences informative for understanding how LLMs learn, when to trust outputs, and paths to improve performance.
>
> The issue with Luo et al. (2024a) is a training confound, not prompting. Their inconsistent BOS token and tokenizer means the backward model saw different token sequences than the forward model, invalidating conclusions about training direction. Valid comparison requires token order as the only variable.
>
> Consider the training sequence "The cat sat on the mat." A forward-trained model learns: P(The|BOS) × P(cat|The) × P(sat|The, cat) × P(on|The, cat, sat) × P(the|The, cat, sat, on) × P(mat|The, cat, sat, on, the). A backward-trained model learns: P(mat|BOS) × P(the|mat) × P(on|mat, the) × P(sat|mat, the, on) × P(cat|mat, the, on, sat) × P(The|mat, the, on, sat, cat). Both models approximate the complete sequence probability.
>
> However, without consistent BOS tokens, models diverge: forward models exclude first token "The" while backward models exclude last token "mat"; no longer approximate the same sequence (Introduction; changes highlighted).
>
> This point is critical to Kalini et al.'s central claim. If forward language is "easier" than backward language, models must train on identical data with token order as the sole variable. Without consistent BOS tokens, two models don't approximate the same probabilities, invalidating comparisons. We've clarified these criticisms are limited to establishing formal equivalence (Introduction; changes highlighted).
>
> While BOS tokens aren't mathematically required, autoregressive transformers need non-empty starting tokens for next-token prediction. Without consistent BOS tokens, models use sequence-first tokens as starting points, meaning forward and backward models don't approximate the same probabilities even with identical data.
>
> We thank the reviewer for recognizing that our work will interest the TMLR audience. We hope after reading our paper that the proof is unsurprising and clear! Indeed, prior to working through these issues, it was surprising to us and others as evidenced by the work we cite that doesn’t get the issue exactly right. We agree that the points we make through proof have come up in related contexts. Given the disagreements we have had on the formal equivalence of different orders, we think it’s a good idea to publish this proof in TMLR and make clear its generality.
>
> Our proof clarifies the situation and guides proper model evaluations. Through extensive experiments, we identified positional bias in masked self-attention as a key factor, complementing research on architectural bias and attention sink phenomena. Our theory-driven methodology revealed flaws in existing work—many studies mentioned theoretical equivalence without formal proofs, leading to experimental setups deviating from theoretical foundations compromising scientific rigor. We hope spelling out these methodological issues will positively affect other research.
>
> We thank the reviewer for their suggested changes, which prompted us to better articulate our central critique. In our revision, we have now clarified that all of the mentioned existing works center on comparing models trained on different orderings and making claims about whether forward or backward language is in some way special or interesting (changes highlighted on Page 2).
>
> While these works make valuable contributions to understanding language model behavior, our analysis focuses on a different but foundational issue. These studies all rely on comparing models trained on sequences with different token orderings to support their claims about the relative merits of different training approaches. For such comparisons to yield valid conclusions about the effects of token ordering, the ordering must be the sole variable that differs between models - a principle rooted in the mathematical equivalence of joint probabilities. Our work identifies methodological considerations in these studies that may affect the validity of such comparisons. While choices such as retraining tokenizers may represent reasonable practices in other research contexts, they introduce additional variables that complicate claims specifically about token ordering effects. We have edited the Introduction (changes highlighted) to enhance the contribution.

---

### Review · Reviewer_pj9v · 2025-09-17

**Summary Of Contributions:**

The paper „Probability Consistency in Large Language Models: Theoretical Foundations Meet Empirical Discrepancies” looks at the behavior of auto-regressive, decoder-only LLMs when trained with different token orderings, i.e., forward order, backward order and random permutations. The paper starts with a proof that the perplexity is independent of the token order, including permutations, which extends prior results that only considered forward and backward order. The authors then report empirical evidence for GPT-2 models of different sizes, trained with different token orders. The authors find that there are only small differences between forward and backward training that are not significant for the benchmark task studied. Permutation leads to big differences, which the authors state does not agree with the underlying theory.

**Audience:**

No

**Audience Explanation:**

The proof and empirical study seems to be based on a flawed assumption about LLMs.

**Claims And Evidence:**

No

**Claims Explanation:**

I believe that authors ignore a crucial aspect in their study: they are dealing with unidirectional language models, which means that probabilities only depend on the text that was already seen. Given how semantics in language works, if random permutations would lead to similar, unidirectional models, it does not make sense to me.

This is also why the proof might be formally correct, but practically irrelevant: The probability distributions in the equation in Line 2 on Page 4 on the right-hand side and left-hand side are not the same, they just look the same due to shorthand notation we commonly use. The full form would be:

$P_{X_0, X_{\sigma(1)}, …, X_{\sigma(n)}}( X_0, X_{\sigma(1)}, …, X_{\sigma(n) = P_{X_0, X_1, … ,X_n}(X_0, X_1, …, X_n}$

This shows that the core assumption of the whole proof is that the joint probability should be the same, independent of the token order – which is clearly not the case for auto-regressive models that model probabilities in a unidirectional manner.

Consequently, unless I miss something crucial, I see no reason why auto-regressive models should show any reasonable patterns or patterns that are similar when tokens are randomly permutated.

(Note: One could make a stronger case for bi-directional models, but this would still assume that the relative distance between tokens and their absolute positions are encoded in such a way, that the token order is not a factor – an assumption that my hold for learned embeddings that where I also see no why to make this work for something like RoPE. Anyways, since no such models are considered here, this is beside the point.)

**Requested Changes:**

Clarify why order of tokens should be irrelevant for unidirectional LLMs and why the core assumption of the proof that the underlying random process that is modeled by a autoregressive, decoder-only transformer model should be order-independent.

---

> ### Author Response · Authors · 2025-09-24
> **Response to reviewer comments**
>
> We thank the reviewer for engaging with our work. To be clear, we appreciate that auto-regressive LLMs are directional – all conditional probabilities are calculated with respect to the preceding context, not future tokens. One of our key observations is that the conditional probabilities associated with that preceding context factorize the joint probability distribution and that the joint probability distribution theoretically should be the same no matter the fixed token order (e.g., forward, backward, or other) an LLM is trained and evaluated on. Thus, when models trained and evaluated on their fixed order deviate in practice from other models operating on different orders, it is indicative of an inconsistency in the conditional probabilities of one or both models. Why these inconsistencies arise in practice is an important question we touch on that has both theoretical and practical implications for understanding how LLMs learn, determining when should we trust their outputs, and opening paths to improve their performance. Our empirical investigation into these practical inconsistencies revealed a positional bias in self-attention, where attention weights disproportionately favor both nearby and very distant tokens. This discovery complements existing research on phenomena like "attention sinks" and the "lost-in-the-middle" problem, and provides a concrete architectural explanation for the observed deviations in model behavior.
>
>
> We have revised our manuscript (toward the end of Sec. 2; changes highlighted) to make it clearer that each auto-regressive LLM captures only a single factorization of the training data. For example, it is not the case that an auto-regressive LLM trained on a forward ordering could be evaluated in a sensible fashion in a reverse token ordering. This is not our claim, nor is it germane to our proof. Instead, each LLM needs to be trained and evaluated on a particular token ordering (e.g., factorization of the joint probability distribution).
>
>
> We have also revised the manuscript (Discussion; changes highlighted) to better explain why it is counterintuitive, yet true, that natural language (like any other sequence of tokens) has the same perplexity for any fixed token ordering. Of course, there is no natural language on earth that has this structure and for good reason for which we will discuss. Besides locality and other biases (e.g., constant bit rate), generation (as opposed to evaluating the perplexity of an existing passage) prove challenging under arbitrary (but fixed) permutations.
>
>
> To provide further detail, we view randomly permuted sequences (with fixed randomness within the context window) as a generalization of backward-trained models where training sequences undergo string reversal. When training GPT-2 on permuted sequences, we still optimize the autoregressive loss by predicting the next token from observed tokens, making this essentially equivalent to training GPT-2 on forward or backward text. Notably, GPT-2 uses learnable absolute position embeddings (which we also train from scratch), so the model sees no inherent ordering and must learn from the fixed permuted sequence through next-token prediction.

---

### Review · Reviewer_72wi · 2025-10-08

**Summary Of Contributions:**

This work provides a theoretical framework for the lens of probability to interpret the autogressive training of LLMs. It also aligns with some empirical studies to connect the information-theoretical aspects (namely Perplexity Invariance) and the behaviors of LLM training on real data.

The theoretical results proved that sequence perplexity, which measures the joint probability of a sequence, is invariant to token orderings (e.g., forward, backward, or arbitrary permutations) under the chain rule of probability. This measure establishes a theoretical benchmark for evaluating the capability of LLM to learn consistent probability distributions. In addition, the results also demonstrated that the choice of token prediction order during training should not affect perplexity if the model accurately captures conditional probabilities. However, practical LLMs approximate probabilities and are trained on specific factorizations, limiting their ability to model other token orderings.

The empirical studies were carried out using GPT-2 at varying sizes (but less than 1B) on neuroscience text with forward, backward, and permuted token orderings. The experiments found "deviations" from theoretical invariance: (1) Forward and backward models showed similar perplexities, but forward models consistently performed slightly better; (2) Models trained on permuted text exhibited significantly higher perplexities, diverging from theoretical expectations.

In discussions, authors traced discrepancies to positional biases in self-attention mechanisms. Forward and backward models favored adjacent and long-range tokens. Permuted models displayed disrupted attention patterns, leading to higher entropy and distinct learning dynamics. However, the theoretical aspects of this work do not propose new theorems or groundbreaking propositions but rather formalize existing probability principles (e.g., the chain rule of probability) to establish a benchmark for evaluating perplexity invariance across token orderings. The theoretical and experimental components are loosely connected in this paper, with the experiments primarily serving to highlight discrepancies between theoretical expectations and practical implementations.

**Additional Comments:**

N/A

**Audience:**

Yes

**Audience Explanation:**

This work intends to discover some information-theoretical foundations of LLM training.

**Claims And Evidence:**

No

**Claims Explanation:**

The central thesis of this paper could be stated more explicitly. It's unclear whether the primary contribution is the theoretical formalization of perplexity invariance or the empirical discovery that current models fail to achieve it due to architectural biases. While the theoretical work formalizes established principles, the experiments compellingly highlight a significant gap between theoretical expectations and practical model behavior. I suggest the authors reframe the narrative to more clearly foreground the empirical diagnosis of model limitations as the core discovery of the work, as this appears to be its most significant contribution.

Maybe I wrong. But it looks like that the paper proposes a *new way* to interpret LLM training by leveraging the classic chain rule of probability. The *new way* here means nobody did it rather than novel techniques. Yet, experiments on smaller models show this theoretical ideal breaks down in practice (models trained using permuted text), though in some cases theortical investigation and empirical observations were aligned.

**Requested Changes:**

1. Please clarify the novelty of the theoretical framework. Could you reduce your theoretical investigation to characterize any asymptotic or non-asymptotic behaviors of LLM training subject to params, computes, data distribution characteristics of training tasks (e.g., ergodicity, if the training sequence is ergodic?), and etc.? Maybe the limits/bounds of perplexity invariance under certain quantitative assumptions on training sequences would be more interested in.

2. If it is impossible to improve the theoretical results. Please emphasize that the paper does not propose new results but rather applies the chain rule of probability in a novel way to interpret LLM training. This distinction helps set realistic expectations for the theoretical contribution. However, in this way, you have to identify why the divergence between theoretical measure of perplexity invarance and experiment results are critical or important. Otherwise, I cannot see the motivation of this work.

3. Please improve the connections between theoretical investigation and empirical studies. Explicitly state that the experiments validate the theoretical framework in some cases (e.g., forward vs. backward training) while revealing its limitations in others (e.g., permuted training).
Discuss how the theoretical framework can guide future research into mitigating the architectural biases that cause these deviations.

---

> ### Author Response · Authors · 2025-10-19
> **Response to reviewer comments**
>
> We thank the reviewer for the thoughtful comments. We view our main contribution is not one or the other, but their integration: theory-informed LLM training evaluations. The theoretical formalization of perplexity invariance directly motivated our empirical design—it prompted us to train GPT-2 models across different token orderings and revealed design flaws in prior work (e.g., inconsistent BOS token usage, varying tokenizers). Conversely, the experiments uncovered practical explanations for theory-practice gaps, which justified formalizing the invariance property in the first place. The contribution lies precisely in this reciprocal relationship: theory guided our experimental design, and experiments revealed why practitioners fail to achieve theoretical expectations. This synergy, rather than a primary focus on either component alone, is what the paper aims to convey.
>
> However, we agree with the reviewer on the need to clarify that the chain rule itself is not a novel theoretical piece; instead, the novelty lies in trying to interpret LLM training via this formal framework, which as we explained above, is an integrated contribution. We have made this clear in the revised paper (changes highlighted toward the end of first paragraph on page 2).
>
> We appreciate the reviewer pushing us to clarify the theory-practice connections. To first address a point of clarification: our experiments do show a statistically significant difference between forward and backward cases, though notably smaller than the permuted training comparison. This itself is revealing—the divergence from theory exists even in this closer case, indicating something systematic is happening. It is also an interesting and perhaps an entire research question as to why the difference is so much smaller compared to permuted training.
>
> We agree with the reviewer in emphasising the importance of understanding the theory-practice gap. In the paper, we have already touched on this in terms of model reliability and interpretability. Specifically, we have discussed that 1) deviations from theoretical probability distributions likely contribute to harmful behaviors like hallucination and unpredictable out-of-distribution performance, 2) systematic positional biases in attention directly linked to phenomena like lost-in-the-middle effects, which undermine model trustworthiness in retrieval and reasoning tasks, 3) models that fail to satisfy basic mathematical principles become fundamentally harder to interpret, predict, and trust in general applications where principled behavior is essential. We have now enhanced the Discussion to emphasise these points further (changes highlighted).
>
> We have now revised the paper (penultimate paragraph of Discussion) to emphasize that we hope this work establishes theoretical groundwork and experimental best practices for investigating divergence from theory. Key questions for future research include: How do these biases scale to real-world scenarios? Do other architectures and positional embeddings exhibit similar patterns? Can we develop better benchmarks for such divergences? What are the downstream implications? Could training-time invariance constraints help close this gap? These directions will be essential for the years ahead, and we are excited to see where this research leads.

---

### Decision · Action_Editor_YGmr · 2025-12-08

**Recommendation:** Reject

**Audience:**

Yes

**Audience Explanation:**

The experimental observations in the paper, especially the explanation of the methodological weaknesses of prior work on measuring the effect of sequence order on LLMs would be interest to some TMLR readers.

**Claims And Evidence:**

No

**Claims Explanation:**

The reviewers appreciated the clarifications and the changes made to the paper by the authors, but they remained unconvinced about some of the claims.

While there were some concerns about the correctness of the proof of "Perplexity Equivalence" (PE), this appears to be just an issue with notation and should be easy to fix. As the Chain Rule states that the joint probability of a vector of random variables can be factorized into per-variable conditionals based on any ordering of the variables, PE is essentially a corollary of it. As a result, just like the Chain Rule, PE holds for all joint distributions, including those defined by LLMs as well as the data distribution models are trained to approximate.

However, PE does not imply that autoregressive models trained on different orderings of the sequence dimensions from the same data distribution should achieve the same perplexity, because each of these models with induce a different joint distribution by capturing the data distribution imperfectly in a different way. Thus it is not at all surprising that the experiments show that training on different data orderings leads to different perplexity values. The reviewers were concerned about this gap between the theory and practice, and it is unclear why the authors concluded that this gap implied that there was something wrong with LLMs rather than with the assumptions of their theory.

The issue here is that the paper makes no distinction between the conditionals in the Chain Rule-based factorizations of the data joint distribution and the conditionals computed/parameterized directly by LLMs trained on sequences from this joint data distribution. PE would apply to the joint distributions learned by different LLMs only if they match the data distribution exactly and thus are identical to the data joint. The authors acknowledge something similar at the end of Section 2, where they state that PE applies to LLMs only if "the model accurately captures the conditional probabilities". However, matching the data conditionals perfectly is simply not a realistic assumption.

With this in mind, it is not clear why it would be desirable for LLMs to satisfy PE in practice, by being equally good at modelling all possible sequence orders. After all, natural languages have strong local dependencies and the fact that LLMs have a good inductive bias for capturing them even if this bias makes them less effective for other orderings seems like a sensible trade-off. Applying PE to n-grams, with and without smoothing, might provide a good way of better understanding the principle's applicability and limitations.